# Unraveling the features of somatic transposition in the *Drosophila* intestine

Katarzyna Siudeja[1,2,*,†] iD, Marius van den Beek[1,2,†] iD, Nick Riddiford[1,2] iD, Benjamin Boumard[1,2], Annabelle Wurmser[1,2], Marine Stefanutti[1,2], Sonia Lameiras[3] & Allison J Bardin[1,2,**] iD

## Abstract

**Transposable elements (TEs) play a significant role in evolution, contributing to genetic variation. However, TE mobilization in somatic cells is not well understood. Here, we address the prevalence of transposition in a somatic tissue, exploiting the *Drosophila* midgut as a model. Using whole-genome sequencing of *in vivo* clonally expanded gut tissue, we have mapped hundreds of high-confidence somatic TE integration sites genome-wide. We show that somatic retrotransposon insertions are associated with inactivation of the tumor suppressor *Notch*, likely contributing to neoplasia formation. Moreover, applying Oxford Nanopore long-read sequencing technology we provide evidence for tissue-specific differences in retrotransposition. Comparing somatic TE insertional activity with transcriptomic and small RNA sequencing data, we demonstrate that transposon mobility cannot be simply predicted by whole tissue TE expression levels or by small RNA pathway activity. Finally, we reveal that somatic TE insertions in the adult fly intestine are enriched in genic regions and in transcriptionally active chromatin. Together, our findings provide clear evidence of ongoing somatic transposition in *Drosophila* and delineate previously unknown features underlying somatic TE mobility *in vivo*.**

**Keywords** *Drosophila* midgut; somatic genome; transposable elements
**Subject Categories** Chromatin, Transcription & Genomics; Development
**The EMBO Journal (2021) 40: e106388**

## Introduction

Transposable elements (TEs) are DNA sequences that shape evolution through their capacity to amplify and mobilize, thereby altering the structural and regulatory landscape of the genome. Numerous mechanisms restrict the mobility of TEs and therefore their mutagenic potential. In germline and somatic cells, TE silencing is achieved by chromatin modifications and small RNA-directed degradation of TE transcripts (Molaro & Malik, 2016; Deniz *et al*, 2019; Cosby *et al*, 2019). The escape of TEs from silencing allows their propagation in the genome. While *de novo* TE insertions in the germline are relatively easy to detect as they result in heritable genomic changes that can be detected through sequencing, TE mobility in somatic cells is more difficult to study. Indeed, the heterogeneity of transposition events within somatic tissues imposes technical challenges as rare TE insertion events affecting a subpopulation of cells often fall below the limits of detection. Thus, the degree to which TEs evade silencing and contribute to somatic genome alteration is much less well understood in developing and adult tissues.

Nonetheless, evidence for active somatic transposition has been recently mounting. Reporters of transposon activity suggested TE mobility in neuronal lineages in human, mouse, and *Drosophila* (Muotri *et al*, 2005; Coufal *et al*, 2009; Li *et al*, 2013; Macia *et al*, 2017; Chang *et al*, 2019). Additionally, recent use of an engineered *gypsy* retrotransposon trapping cassette in flies suggested that somatic transposition could also occur in non-neuronal tissues such as the fat body (Jones *et al*, 2016; Wood *et al*, 2016) or the intestine (Sousa-Victor *et al*, 2017). Interestingly, increased TE expression in many organisms has been linked to normal tissue aging as well as pathologic conditions of neurodegeneration. Evidence suggests that TE transcription may be linked to disease pathology, however, it remains unknown to what extent TE insertional activity contributes to these phenotypes (Dubnau, 2018; Burns, 2020). Nevertheless, the *gypsy* retrotransposon reporter activity was shown to increase in aging *Drosophila* brain, fat body, and gut (Li *et al*, 2013; Jones *et al*, 2016; Wood *et al*, 2016; Sousa-Victor *et al*, 2017; Chang *et al*, 2019), correlating in some cases with increased DNA damage, and suggesting that TE insertional activity could indeed play a role in age-related deterioration of somatic tissues. However, a major drawback of using engineered reporters is that reporter cassettes could be inactivated by other means than a TE insertion. In addition, the available transgenic lines only report a limited number of TE families. Finally, results obtained with engineered reporters may not necessarily reflect the activity of endogenous elements encoded in the genome.

1 Institut Curie, CNRS, UMR 3215, INSERM U934, Stem Cells and Tissue Homeostasis Group, PSL Research University, Paris, France
2 Sorbonne Universités, UPMC Univ Paris 6, Paris, France
3 ICGex Next-Generation Sequencing Platform, Institut Curie, PSL Research University, Paris, France
  *Corresponding author. Tel: +33 1 56 24 65 62; E-mail: katarzyna.siudeja@curie.fr
  **Corresponding author. Tel: +33 1 56 24 65 80; E-mail: allison.bardin@curie.fr
  †These authors contributed equally to this work

Genomic sequencing has provided some direct evidence for endogenous somatic retrotransposition though it has almost exclusively focused on the retrotransposition of LINE1 (L1) elements in human cancers (Lee *et al*, 2012; Solyom *et al*, 2012; Tubio *et al*, 2014; Rodić *et al*, 2015; Doucet-O'Hare *et al*, 2016; Tang *et al*, 2017; Rodriguez-Martin *et al*, 2020) or in human and rodent neuronal tissues (Baillie *et al*, 2011; Evrony *et al*, 2012; Upton *et al*, 2015). However, the first reports of high L1 transposition frequencies in mammalian brains were later shown to be overestimated due to artifacts of sequencing methodology and data analysis (Evrony *et al*, 2016). Similarly, in *Drosophila*, endogenous somatic TE mobility remains controversial as sequencing performed on populations of adult fly neurons failed to identify true insertions among multiple technical artifacts (Perrat *et al*, 2013; Treiber & Waddell, 2017). Thus, the true extent to which diverse classes of TEs affect genomes of somatic tissues remains to be addressed. Moreover, due to low numbers of somatic insertions recovered thus far from non-cancerous conditions, integration site enrichments of TEs in normal tissues *in vivo* are not well understood. Finally, a genetically amenable model system to reliably study somatic transposition is currently lacking.

We have previously established the *Drosophila* midgut as a model system to address the prevalence of somatic mutation in an adult self-renewing tissue (Siudeja *et al*, 2015). The fly midgut is maintained by a population of intestinal stem cells (ISCs) that divide to self-renew and give rise to two differentiated cell types: absorptive enterocytes (ECs) and secretory enteroendocrine cells (EEs) (Micchelli & Perrimon, 2006; Ohlstein & Spradling, 2006). Our previous study demonstrated that ISCs acquire genetic mutations including deletions and complex rearrangements, which have important physiological impact on the tissue (Siudeja *et al*, 2015).

Here, we make use of the fly intestine to demonstrate the contribution of TEs to the somatic genetic variation of an adult tissue. Using whole-genome sequencing of clonally expanded gut neoplasia, we reveal ongoing somatic retrotransposition in the fly midgut. We identify *de novo* TE insertions in the tumor suppressor gene *Notch*, likely contributing to its inactivation and neoplasia formation. Additionally, we apply Oxford Nanopore long-read sequencing of non-clonal healthy adult tissues to provide evidence of tissue-specific differences in retrotransposition. Based on hundreds of high-confidence *de novo* transposition events, we uncover non-random distribution of somatic TE insertion sites in the gut tissue. Transposition occurs throughout the genome and somatic insertions are enriched in genic regions as well as active, enhancer-like chromatin. Overall, by providing direct DNA sequencing-based evidence for *de novo* somatic TE insertions, we uncover novel features of their *in vivo* biology.

## Results

### Somatic TE insertions in the Notch gene identified in spontaneous intestinal neoplasia

We have previously shown that somatic mutations occur frequently in intestinal stem cells (ISCs) and that the spontaneous inactivation of a tumor suppressor *Notch* in male adult ISCs drives the clonal expansion of mutant cells and formation of gut neoplasia (Siudeja

*et al*, 2015). Since *Notch* is located on the X chromosome and as such is present in a single copy in males, a single "hit" can lead to its inactivation (Fig 1A). In contrast, females, harboring two copies of *Notch*, do not or very rarely develop similar spontaneous *Notch* inactivation events. Male neoplasia can be easily distinguished by the clonal accumulation of two intestinal cell types: ISCs expressing Delta and enteroendocrine cells (EEs) marked by Prospero. Our initial sequencing analysis of clonal neoplasia isolated from *ProsGAL4 UAS-2xGFP* (hereafter abbreviated as *Pros > 2xGFP*) male flies, revealed inactivation of *Notch* by large deletions or complex genomic rearrangements (Siudeja *et al*, 2015). In order to expand this analysis and better characterize distinct types of somatic mutations that impact adult ISCs, we generated a large dataset of whole-genome paired-end Illumina sequencing of an additional 30 clonal neoplasia from the same genetic background, as well as four clonal neoplasia from *DeltaGAL4 UAS-nlsGFP* male flies (hereafter abbreviated as *Delta > nlsGFP*), for a total of 37 clonal samples and matched control head DNA sequenced with an average of 47x coverage (Fig 1A and Table EV1). In our analysis, we compared clonal gut samples to their respective head controls to identify somatic events arising only in the gut samples. These data are also analyzed by companion paper that addresses structural variation in the same model system (see Materials and Methods and also preprint: Riddiford *et al*, 2020). As expected, a majority of clonal samples showed evidence for inactivation of the Notch pathway by somatic deletions or complex rearrangements (for details see preprint: Riddiford *et al*, 2020). Interestingly, four samples (P15, P47, P51, and D5) did not harbor any other mutation that could explain the clonal expansion, but showed evidence of somatic TE sequence inserted in *Notch* (Fig 1B and C). Due to very limited sample material, we could not perform simultaneous RNA expression analysis in order to directly demonstrate the effect of TE inserts on *Notch* expression. However, as we did not detect evidence of other genetic alteration of *Notch* or Notch pathway components, we concluded that the TE insertions were most likely causative of the clonal expansion and *Notch* mutant phenotype. Strikingly, in sample P15, we observed two integrations within *Notch* (Fig 1C), with one of the two events having more sequencing reads supporting the insertion than the other, suggesting that the first insertion inactivated *Notch*, while the second one occurred later during the clonal expansion. All candidate insertions were supported both by clipped reads mapping partially to a TE and partially to *Notch*, and paired-end reads where one mate-pair is TE anchored and the other is mapped to *Notch* (Fig 1B). Among the five candidate insertions identified, three were within the UTR regions of the gene and two TE integrations were in intronic sequences (Fig 1C). For all cases described, no read evidence was found for an insertion in the matched head DNA controls. Thus, TE insertions appeared specific for the clonal gut DNA, suggesting they occurred in somatic gut tissue (Fig 1B).

To validate the *Notch* TE insertions, we designed primer pairs flanking the identified insertion sites and performed a full-length or one-sided PCR amplification using the original genomic DNA as a template (Fig 1D and Appendix Fig S1). Out of the five candidate *Notch* TE insertions, all were PCR validated (4 full-length and 1 one-sided validation). All insertions were amplified only from the clonal neoplastic DNA and not the DNA of matched control tissues from the same fly, confirming that these were true neoplasia-specific somatic TE insertions. Finally, all

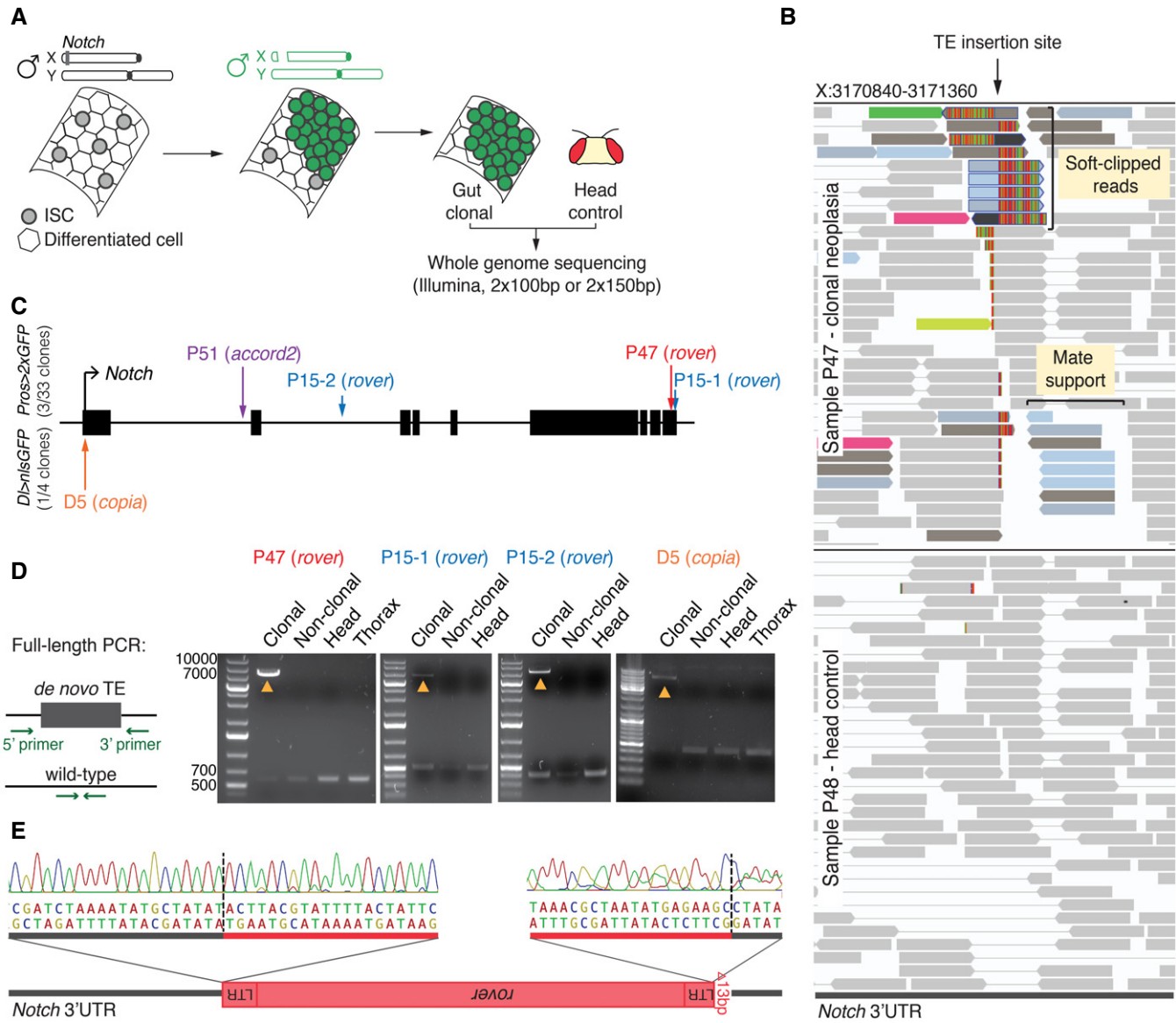

**Figure 1. Somatic TE insertions in *Notch* in spontaneous male neoplasia.**

A   The fly intestine is maintained by the Intestinal Stem Cells (ISCs). In male flies, carrying one X chromosome, the tumor suppressor gene *Notch* is present in a single copy. Inactivation of *Notch* in a stem cell (in green) leads to a clonal expansion of the mutant cell and neoplasia. The neoplastic gut region was microdissected together with the head of the same fly. DNA isolated from both tissues was subjected to whole-genome paired-end sequencing.

B   An Integrative Genomics Viewer (IGV) screenshot of the *Notch de novo* TE insertion site from sample P47 (clonal neoplasia) and its head control, sample P48. Bars represent sequencing reads. Reads supporting the TE insertion are colored according to homology to a specific TE insertion sequence. Multiple colors at a putative insert site frequently indicate homology to different reference copies of the same TE family. Two types of supporting reads can be seen: soft-clipped reads spanning the insertion site and mapping partially to the reference genome and partially to the TE, and mate-pair support reads—flanking the insertion site and mapping to the reference genome but with mates (not seen) mapping to a TE.

C   The *Notch* locus and the identified somatic TE insertion sites indicated with vertical arrows. Black bars represent exons. Insertions in *Notch* were identified in three out of 33 clonal samples from the *Pros > 2xGFP* genetic background and in one out of 4 *Dl > nslGFP* samples.

D   PCR validation of four somatic, neoplasia-specific TE insertions. Primers were designed to target regions flanking the insertion sites. Yellow arrowheads indicate PCR products containing an insertion amplified in the clonal DNA but not in the neighboring gut tissue (non-clonal), head or thorax for the same fly. Short wild-type amplicon was detectable in all samples. Thorax DNA sample was not available for sample P15.

E   Sanger sequencing of the TE insertion breakpoints in the 3'UTR of *Notch* from sample P47. The *rover* LTR element was inserted in a reverse orientation to *Notch*. The 5' LTR sequence was truncated by 13 bp. Vertical dashed lines indicate insertion breakpoints. LTR—long terminal repeat.

insertions were partially or fully sequenced-verified by Sanger sequencing (Fig 1E and Appendix Fig S1).

Altogether, these data revealed that TEs actively transpose in the adult midguts. Importantly, TEs can insert into the *Notch* tumor suppressor gene in stem cells, likely driving neoplastic growth in male flies.

## Retrotransposition occurs genome-wide in the fly midgut

Having identified that TEs are mobile in the fly midgut and likely inactivate the *Notch* locus, we then aimed to address the prevalence of somatic transposition on a genome-wide scale. To precisely map somatic TE insertions from our short-read sequencing data, we developed a dedicated pipeline (Fig 2A, details in Materials and Methods) and applied it to neoplastic and matched control samples.

For further analysis, we retained only insertions bearing a target site duplication (TSD) as a footprint of transposition-dependent events. TSDs are short, identical, duplicated sequences generated on both sides of a TE insertion as a consequence of a staggered endonuclease cut of the target DNA (Feng *et al*, 1996). We identified a total of 674 (median of 15 per clonal genome) somatic insertions with TSDs from the *Pros > 2xGFP* background and 97 (median of 23 per clonal genome) integrations in the *Delta > nlsGFP* samples, all of which were private to gut clonally amplified samples and not present in the matched control DNA, or any of the controls (Fig 2B, Table EV2). In both genetic backgrounds, a great majority of identified insertions were retrotransposons (Fig 2C), suggesting that this TE class is the most mobile in the gut tissue. In the *Pros > 2xGFP* background, the most abundant were insertions of *rover* elements (487 insertions), followed by *copia* (102 insertions), *diver* (7 insertions), *blood* (5 insertions), *roo* (4 insertions), and sporadic insertions of other LTR TE families (Fig 2C). Among non-LTR retroelements, we identified *de novo* integrations of LINE-like retrotransposons, including 32 *de novo* insertions of *I-elements*. Insertions of terminal inverted repeat (TIR) DNA elements and *foldback* elements were infrequent (Fig 2C). In the *Delta > nlsGFP* background, we mapped 16 insertions of *copia* elements, 10 *roo* integrations, followed by *297* (9 insertions), *opus* (8 insertions), *mdg1* and *Tabor* (7 insertions each), and other LTR TE families (Fig 2C). Rare integrations of LINE-like elements and DNA TIR class TEs were also found. Although we observed varying levels of transposition, there were no striking differences in the types of mobile TEs between samples of the same genetic background, suggesting that active TEs did not differ greatly between individuals (Fig EV1). In contrast, differences in mobile TE families were evident between the two genotypes, suggesting that the repertoire of somatically mobile TEs likely depends on the genetic background. However, we cannot exclude that some observed differences in mobility may have resulted from the differences in cell-type-specific clone composition between the two genotypes, with either an enrichment of enteroendocrine cells (*ProsGal4* driven GFP) or intestinal stem cells (*DeltaGal4*-driven GFP).

To further confirm whether the identified TE insertions were indeed true transposition events, we analyzed somatic TSDs for all TE families which produced at least six *de novo* insertions and compared these with known germline TSDs. Most LTR elements generated short TSDs with a median length below 10 base-pairs (5 bp for *rover*, *copia*, *roo*, *297* and *diver*; and 3 bp for *blood*), consistent with TSD lengths reported previously for germline insertions of LTR elements (Dunsmuir *et al*, 1980; Linheiro & Bergman, 2012) (Fig 2D). Three LTR elements, *opus*, *Tabor* and *mdg1*, produced unexpectedly long TSDs with a median of 23, 26, and 25 bp, respectively, in contrast to 4 bp reported previously (Linheiro & Bergman, 2012). However, with relatively low numbers of somatic insertions of these TE families, it is difficult to conclude if this discrepancy with previously published reports could be

biologically relevant. TSDs generated by LINE-like elements were, in general, less strictly defined but centered above 10bp (median of 12, 25, and 11 for *I-*, *F-*, and *Doc-elements*, respectively, Fig 2D), in agreement with previous reports (Bucheton *et al*, 1984; Sang *et al*, 1984; Driver *et al*, 1989; Berezikov *et al*, 2000). Finally, we searched for target site motifs of the most represented TEs. A highly significant (AT)-rich target site sequence motif around insertions sites was identified for the *rover* LTR element reflecting non-random integration (Fig 2E). Although there are no previous reports about target site preferences of *rover* elements, TEs from closely related classes (such as *297 or 17.6*) show similar (AT)-rich target motives (Whalen & Grigliatti, 1998; Bowen & McDonald, 2001; Linheiro & Bergman, 2012). The second most mobile element in our datasets, *copia*, did not show target site preference, which is consistent with previous reports from germline analyses (Dunsmuir *et al*, 1980). Altogether, our data show that genome-wide somatic TE integration sites have similar characteristics to germline insertions. This lends further support to the detected TE insertions in the gut being true somatic transposition events, rather than random DNA integrations or products of chimeric reads.

Notably, using our detection criteria, we identified only rare somatic TE insertions in the head samples of both genotypes sequenced (median of 2 insertion/sample in *Pros > 2xGFP* heads and 2.5 insertions/sample in *Dl > nlsGFP* heads, Table EV3 and Appendix Fig S2). However, the frequency of transposition between gut and head samples cannot be directly compared in this assay. Indeed, the head is a heterogeneous cell population, and therefore, somatic transposition in a few cells of the head would be below the detection level in our analyses. In contrast, the intestinal neoplasia are clonal expansions of single ISC genomes, increasing likelihood of detecting TE insertions. Accordingly, the rare somatic insertions identified in head samples had only a few clipped and mate-pair supporting reads, reflecting that these were likely rare events present in limited numbers of cells (Appendix Fig S2C). This difficulty to detect TE insertions in non-clonal fly head DNA is also in agreement with recently published data (Treiber & Waddell, 2017). Because single cell insertions are unlikely to be detectable in our assay, we believe that the identified head insertions probably occurred during brain development leading to a small clone of cells harboring the TE insert, rather than in an adult fly brain, which is post-mitotic. Alternatively, they could represent rare but recurrent insertions arising independently in multiple cells of the adult fly brain.

Overall, we conclude that somatic retrotransposition in the fly midgut is not limited to the *Notch* locus, but occurs genome-wide. LTR elements are the most active, while LINE-like retrotransposons mobilize less frequently. Although TE families identified as the most mobile can differ between fly strains, our data suggest that retrotransposons are frequently active in gut tissue.

## TE insertions arise before and after the clonal expansion

To better understand when somatic transposition occurs in the fly gut, we then used allele frequencies to estimate the timing of genome-wide *de novo* integrations identified in clonal samples relative to the event inactivating *Notch* and initiating the clonal expansion (Fig 3A). The allele frequency is the ratio of sequencing reads supporting and opposing any given insertion. Assuming the observed allele frequency represents the true allele frequency in the cell population,

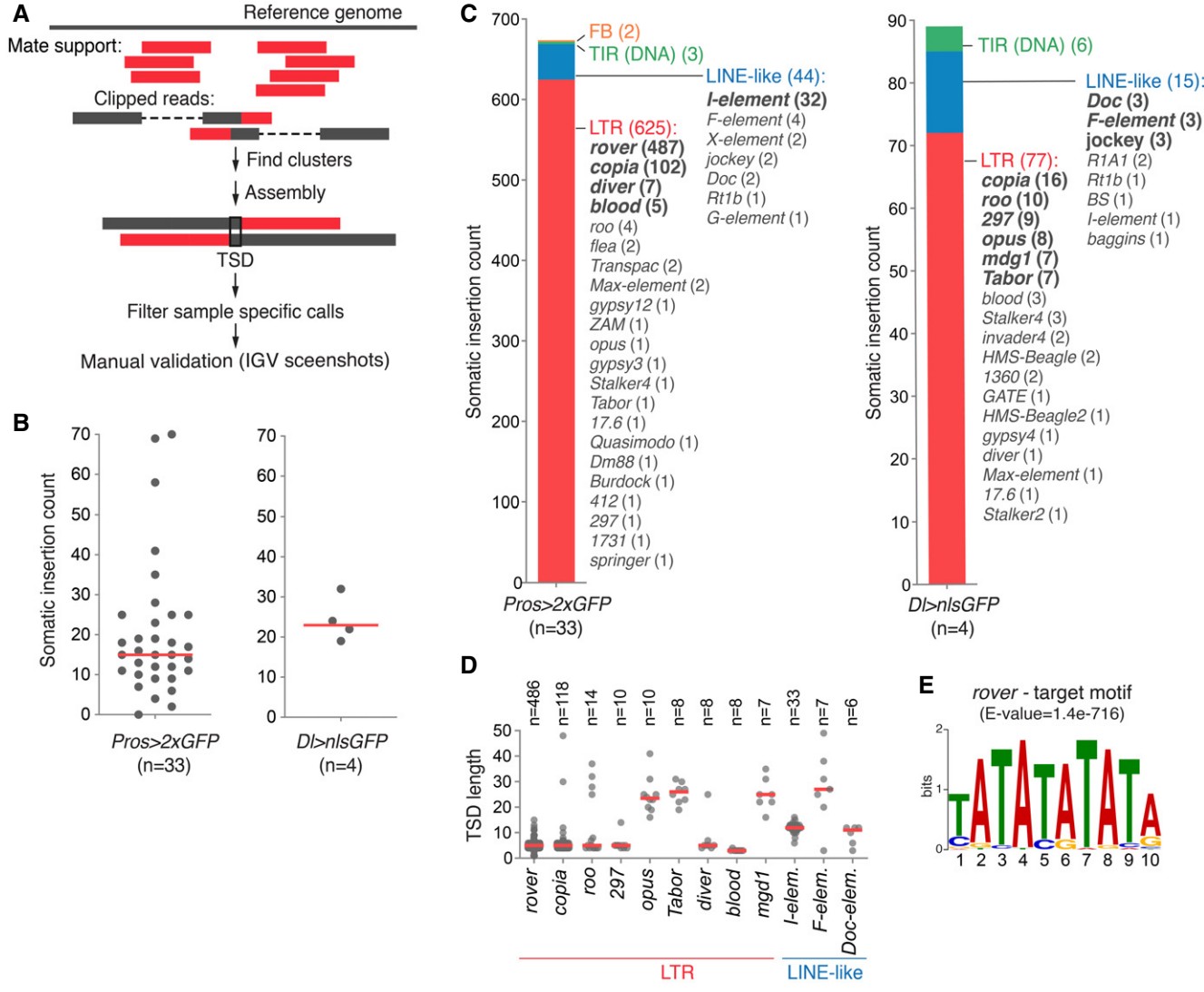

**Figure 2. Retrotransposition occurs genome-wide in the fly midgut.**

A The bioinformatic pipeline used to identify somatic TE insertions in short-read sequencing datasets. Two types of supporting reads are identified genome-wide: mate support reads, where one of the paired-end reads is mapped to the reference genome, while the other mate (not shown) is associated with a TE, and clipped reads, which span the insertion site and map partially to the reference genome and partially to a TE. Isolated reads were then clustered and assembled to map individual insertion sites. Only insertions with a valid target site duplication (TSD) were retained, sample-specific calls were filtered, and manual validation of each candidate insertion was performed on IGV.

B The frequency of gut-specific somatic insertion sites in the *Pros > 2xGFP* and *Delta > nlsGFP* genetic backgrounds.

C The distribution of TE classes active in the two genetic backgrounds studied. TEs were categorized in four main classes: LTR—long terminal repeat retrotransposons, LINE-like—non-LTR retrotransposons, TIR—terminal inverted repeat DNA transposons, and FB—*foldback* element.

D TSD length distribution for somatic insertions of most frequent TE families. Insertions from both genotypes were pooled.

E The target site motif found around (± 10bp) *rover* LTR insertion sites recovered from the clonal gut samples. (E-value was calculated with MEME (Bailey *et al*, 2009), where E-value < 0.05 is considered statistically significant).

Data information: In (B and D), red lines represent median values.
Source data are available online for this figure.

it can be used as an estimate of transposition timing. A TE insertion could arise before the onset of neoplasia, either during development or in the young adult gut, and be present in some cells of the normal tissue. Upon the inactivation of *Notch*, a stem cell would initiate clonal expansion and, at the time of analysis, the insertion would be present in all clonal cells as well as neighboring "normal" cells isolated for sequencing along with the clone. Such a variant would present estimated allele frequency equal or greater than the *Notch* mutation. Alternatively, transposition could also occur after the clonal expansion, in which case such insertion would then be present in a fraction of cells of the clone and show estimated allele frequency lower than that of the Notch pathway inactivating mutation.

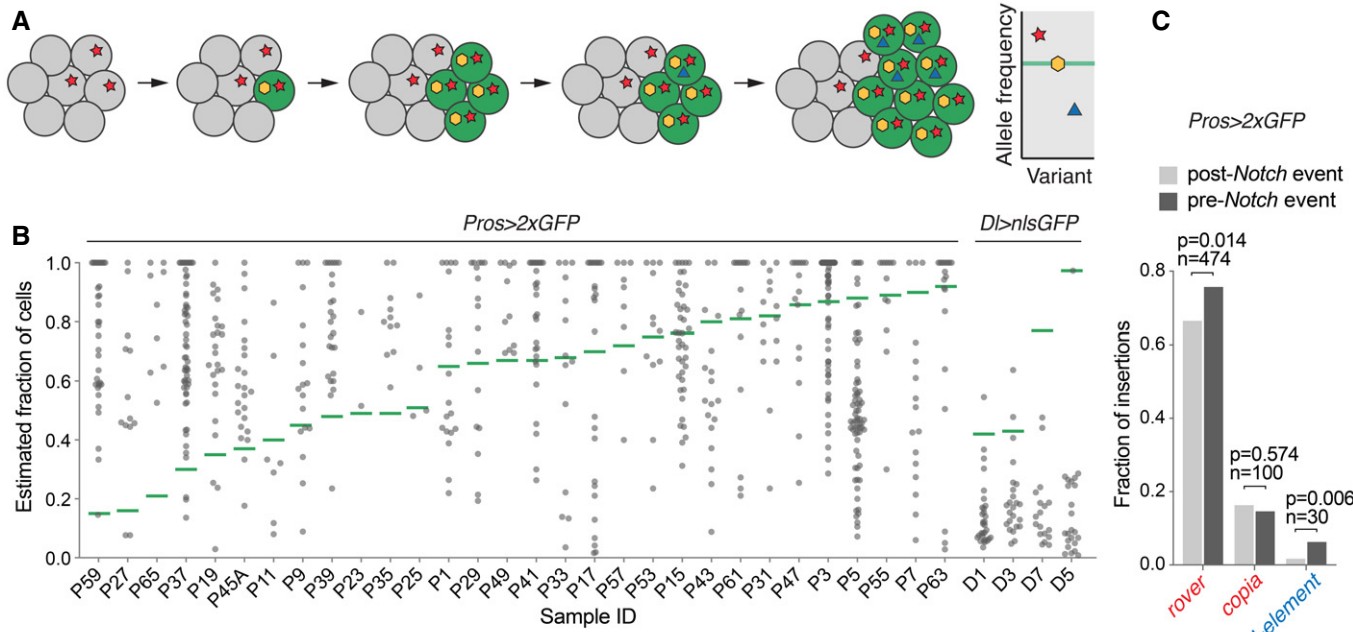

**Figure 3. Somatic retrotransposition occurs before and after the clonal expansion.**

A   A somatic insertion may arise in a normal tissue and be present in a fraction of cells of the tissue (red star). If a second somatic event (yellow) inactivates the Notch pathway, the mutant cell (indicated in green) will initiate the clonal expansion amplifying somatic variants already present in its genome. Finally, any insertion that occurs after the clonal expansion (blue triangle) will be present in a subset of neoplastic cells. The graph represents theoretical allele frequencies of somatic variants obtained from the sequencing of clonal tissue samples. The allele frequency of a *Notch*-inactivating event, marking the onset of neoplasia is represented with a green horizontal line. A somatic insertion with allele frequency higher than the *Notch*-inactivating event was likely present in the tissue before the clonal expansion. In contrast, an insertion with allele frequency lower than the *Notch*-inactivating event likely occurred after the initiation of neoplasia and is thus subclonal.

B   The fractions of cells containing an insertion for all somatic TE insertions identified in neoplastic samples were estimated based on calculated variant allele frequencies (see Materials and Methods). The onset of neoplasia is represented with a green horizontal line. Three samples (P45B, P51, and P21), where timing of the neoplasia onset could not be unambiguously estimated, were excluded from this analysis.

C   The distribution of somatic insertions with estimated cell fraction higher or lower than *Notch*-inactivating mutations for the most represented LTR elements (*rover* and *copia*) and LINE-like *I*-elements (*P*-values were calculated with Fisher's exact test, two-tailed, n = number of insertions).

Source data are available online for this figure.

In both genetic backgrounds sequenced, we uncovered TE insertions in the *Notch* locus likely driving neoplasia formation, indicating pre-clonal TE mobility (Fig 1C). In the *Pros > 2xGFP* background 63.5% of genome-wide somatic integrations showed allele frequency higher than the *Notch* pathway inactivating event in the same sample (Fig 3B). These insertions were thus likely present in the tissue before the onset of neoplasia and occurred either during gut lineage specification in development or in young adult life. The remaining insertions (36.5%) were of lower allele frequency than *Notch* pathway inactivating events, indicating that transposition continued to occur within clonal populations of cells after the initiation of a neoplasia (Fig 3B). High- and low-allele frequency insertions were detected for all TE families for which insertion counts were high enough to allow such analysis (Fig 3C). The LTR elements *rover* and *copia* inserted equally often before and after the neoplasia-initiating event. In contrast, LINE-like *I-element* integrations were moderately, but significantly, enriched among variants with allele frequencies higher than the *Notch* events, suggesting that this TE family might be more active during development in the precursor lineage of the gut or in the normal adult guts prior to the onset of neoplasia. In contrast to *Pros > 2xGFP* neoplastic clones, insertions in *Delta > nlsGFP* samples were largely

subclonal to the neoplasia-initiating events (96.8%), indicating that in this genetic background a majority of detected TE integrations occurred in the adult gut after the onset of neoplasia (Fig 3B). However, the few remaining insertions, including one in *Notch*, likely causative of *Notch* inactivation, indicate that pre-clonal mobility also occurred in this genetic background.

Together, these results imply that somatic retrotransposition in the fly gut can drive inactivation of a tumor suppressor *Notch* and that it can occur before and after the clonal expansion. This suggests that retrotransposon activity is not restricted to neoplasia and can act through adult life in the gut and perhaps also during development.

### TE expression levels do not predict their mobility

The expression and activity of LINE1 (L1) elements, the only somatically active TEs identified to date in the human genome, are increased in different tumor types (reviewed in Burns 2017). We wished to address whether the initiation and clonal expansion of gut neoplasia could lead to TE deregulation. Thus, we asked if inactivation of *Notch* in a stem cell, which leads to a clonal neoplasia, could also cause increased TE expression. To do this, we compared TE expression levels in previously published RNA sequencing data of FACS-sorted

wild-type and *Notch* RNAi knockdown ISCs (Patel *et al*, 2015). Expression of most TEs was not affected upon *Notch* knockdown (Appendix Fig S3). The few TE families that were significantly differentially expressed did not overlap with the mobile TE classes identified in our assays (Appendix Fig S3). Even though this data set comes from a different genetic background than the one we used to isolate neoplastic clones, the results suggest that neither inactivation of the Notch pathway nor hyperproliferation of the gut tissue are sufficient to strongly deregulate TE expression.

We next asked whether there was a correlation between TE expression levels and their mobility. To do so, we performed RNA sequencing of normal (non-neoplastic) *Pros > 2xGFP* midguts and compared this with our data on *de novo* TE insertions (Fig 4A). Most TEs showed very low (TPM, transcript per million < 1) or low (1 < TPM < 5) transcript levels. The most highly expressed elements, *copia*-LTRs (TPM = 481), were also the second most mobile in this genetic background. However, mobility did not directly correlate with TE expression levels, as the most active LTR elements *rover* and the most active LINE-like *I-elements*, were both very lowly expressed in the tissue (TPM < 1) (Fig 4A). For certain moderately expressed TE families, such as LTR element, *invader1*, or the LINE-like element, *Juan*, the canonical TE sequence was only partially covered, suggesting that full-length, transposition-competent copies were not transcribed (Fig 4B). Notably, these data show that, at the tissue-wide scale, steady-state levels of TE transcripts are not good predictors of TE mobility.

On the post-transcriptional level, somatic control of TEs is mostly achieved by the endogenous siRNA (endo-siRNA) pathway (Chung *et al*, 2008; Czech *et al*, 2008; Ghildiyal *et al*, 2008). Our gut transcriptome analysis showed that siRNA pathway genes, *Argonaute 2* (*AGO2*), *Dicer-2* (*Dcr-2*), *loquacious* (*loqs*), and *r2d2* were expressed, suggesting that this pathway is functional in the fly midgut (Fig 4C). This was further confirmed by sequencing of the gut small RNA fraction, which detected short 21-nucleotide sense and antisense reads complementary to TEs, as expected for the *Drosophila* siRNAs (Fig 4D). We found that siRNA levels were not directly proportional to the TE transcript levels (Fig 4E). The five active TE families (*rover*, *copia*, *I*-element, *diver* and *F*-element), responsible for 93.8 % of all insertions, showed low levels of 21-nt antisense siRNAs. Thus, post-transcriptional silencing by siRNAs of these elements could be inefficient. Nevertheless, low siRNA levels were not a prerequisite for mobility, as we also detected *de novo* insertions of TEs (including *blood* and *roo*) that had abundant siRNAs present (Fig 4D and E). Importantly, this implies that low levels of siRNAs could allow for the somatic mobility of some TEs, while other TEs retain their ability to mobilize even in the presence of abundant siRNAs.

Recent reports suggested that the PIWI-interacting RNA (piRNA) pathway, known to control TEs in the gonads (Brennecke *et al*, 2008; Chambeyron *et al*, 2008), could also play a role in somatic TE silencing (Perrat *et al*, 2013; Jones *et al*, 2016; Sousa-Victor *et al*, 2017). However, it remains to be proven whether piRNAs are indeed produced in somatic tissues. We did not detect abundant 23-30-nucleotide-long RNAs (characteristic of piRNAs) in the analyzed gut small RNA samples (Fig 4D). Thus, if piRNAs are produced in the gut, they are at low levels and were under our detection limit. In contrast, 23-30-nucleotide-long RNAs with a typical "ping-pong" signature, indicative of piRNAs (Brennecke *et al*, 2007;

Gunawardane *et al*, 2007), were easily detected in ovary controls of the *Pros > 2xGFP* females. These were complementary to all TEs, including somatically active TE families (*rover*, *copia*, and *I*-element), suggesting that piRNA-mediated TE silencing of these TEs was properly established in the female germline (Fig EV2A and B). There were also no significant differences in ovary piRNA levels between two parental stocks (*ProsGAL4* and *UAS*-2xGFP) used to obtain the *Pros > 2xGFP* flies (Fig EV2C and D). Thus, the observed somatic TE activity could not be explained by differences in active TEs between the parental genotypes, as previously documented in *Drosophila* dysgenic crosses (Brennecke *et al*, 2008).

Altogether, we find that in the fly gut, neoplastic transformation is not necessary for TE expression and that many TE families are transcribed in the normal gut tissue. At the tissue-wide scale, TE RNA levels do not correlate with somatic mobility and even very low transcript levels can be sufficient for active transposition. Although post-transcriptional control by the siRNA pathway is in place, some retrotransposons escape this control and mobilize in the tissue.

## Tissue-specific transposition

To further address TE mobility in normal tissues without a clonal expansion, we decided to apply long-read sequencing to bulk genomic DNA obtained from either pooled midguts or pooled heads from the same individuals of the *Pros > 2xGFP* background. High molecular weight genomic DNA was sequenced to 85x coverage using the Oxford Nanopore Technology (ONT). We then detected all full-length, non-reference, and tissue-specific TE copies entirely contained in a sequencing read (Fig 5A). Considering that in the absence of clonal expansion, any somatic insertion would be very rare in sequencing of bulk DNA, we extracted only insertions supported by a single read ("singletons") and classified them as potentially somatic. A possible drawback of such approach could be that a germline population variant present in a single individual could be mistaken for a somatic variant. To help to exclude such variants, we eliminated all insertions detected in both gut and head DNA pools. Additionally, we used our short-read clonal datasets to estimate that germline TE variants between individual flies were rare in the *Pros > 2XGFP* background and did not belong the same families that somatically active TEs (Table EV7 and Appendix Fig S4). Among all singleton insertions with a TSD, *rover* LTR elements were detected the most frequently in gut DNA (152 insertions) (Fig 5B and Table EV4). Importantly, the fact that *rover* singleton insertions are the most frequent in both the long-read data as in the Illumina short-read data supports the notion that these are likely somatic *de novo* insertions. Additionally, mapped *rover* insertions found in long-read ONT data had identical AT-rich target site motifs to those identified in clonal samples with Illumina sequencing (Figs 2E and 5C). Similar to the short-read Illumina sequencing, singleton reads from the gut were also found containing other LTR elements (Fig 5B). While a total of 191 singleton insertions, representing putative *de novo* integration events, were recovered from the gut DNA, 24 singletons were also found in the head DNA. This suggests that somatic TE mobility occurs both in the gut and in the head, further supporting our findings of rare inserts in the head from the short-read sequencing data (see Appendix Fig S2). Interestingly, with the exception of singleton insertions from *roo* LTR

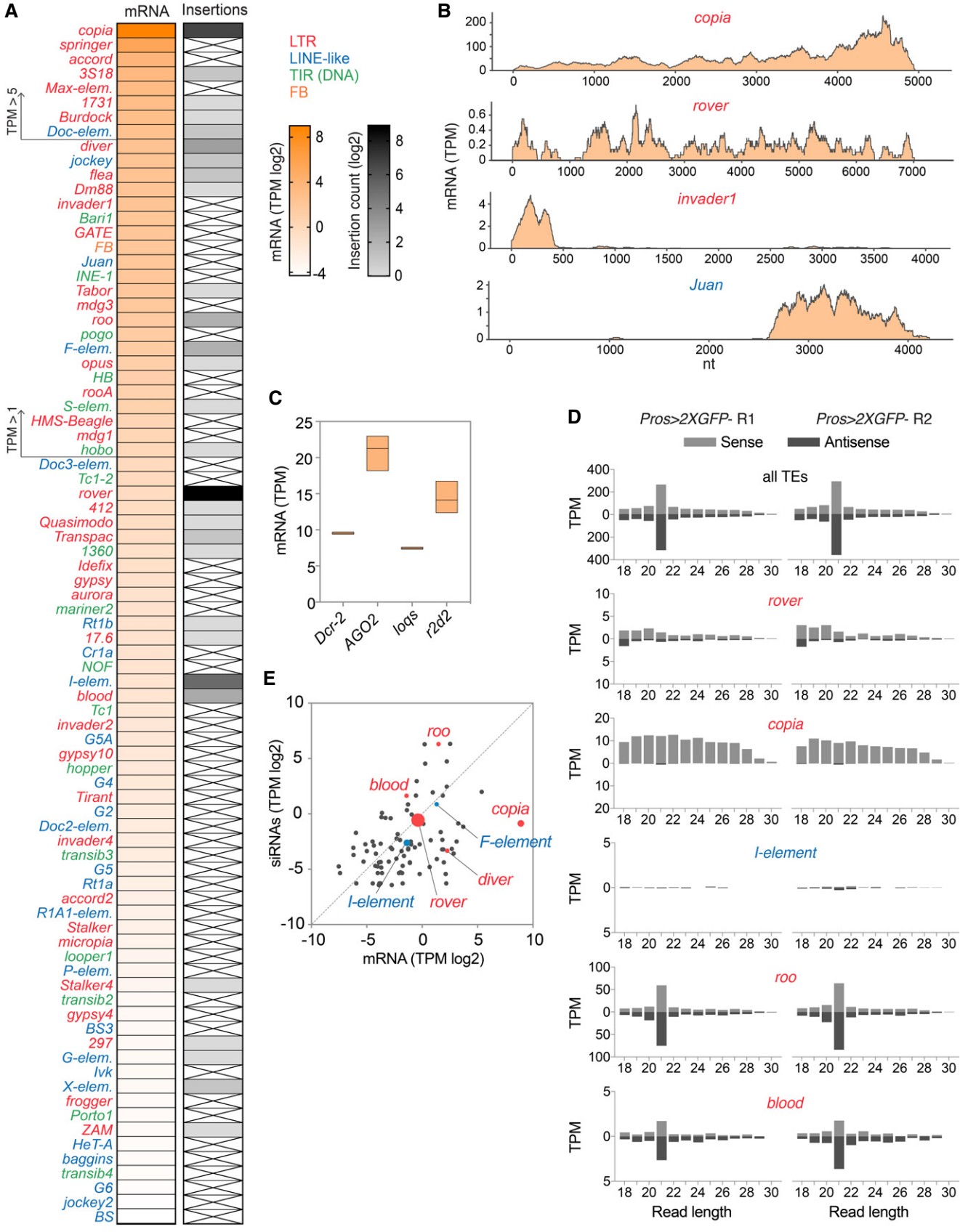

**Figure 4.**

**Figure 4.    TE expression and siRNA pathway activity in the fly midgut.**

A   Heatmaps representing normalized TE expression levels (in log2(TPM), transcripts per million) and mobility (log2(insertion counts)) in *Pros > 2xGFP* midguts. TEs with TPM values below 0.05 (log2(TPM)< −4.3) are not depicted. Crossed out cells represent no somatic insertions of that family identified.
B   Normalized read coverage over the full-length canonical sequence of selected TE families.
C   Normalized expression levels of siRNA pathway genes.
D   The size distribution of sense and antisense reads from gut small RNA fractions mapping to all TEs (upper panel) or selected TE families mobilizing in the gut. R1 and R2 are two biological replicates.
E   Scatter plot of normalized transcript (mRNA) levels and antisense, 21nt siRNA levels for all TE families. Transposons generating somatic insertions are highlighted in red (LTR elements) or blue (LINE-like), with a symbol size reflecting the somatic insertion counts.

Data information: In (C), bars represent the minimum, the maximum, and the mean from three biological replicates. In (D), R1 and R2 are two biological replicates.
Source data are available online for this figure.

elements, almost all singletons from other TE families were specifically found in one of the two tissues, suggesting that TE family activity may be tissue-specific. Singletons from LTR elements *rover*, *springer* and *copia*, contributing together 82.8 % of all putative somatic insertions, were found only in the gut tissue. Singletons of other elements (such as *Stalker2*), although rarer, appeared specific to the head DNA (Fig 5B). This suggests that the repertoire of somatically active TEs may differ between different tissues.

To gain further insight into the tissue-specificity of transposition, we then asked if the same TEs were mobile in the germline, leading to heritable *de novo* insertions. To do so, we sequenced 18 individual flies from the progeny of *Pros > 2xGFP* parents along with their respective parents (Fig EV3) and detected *de novo* germline insertions. We discovered three *de novo foldback* element insertions. Of note, we found no germline *de novo* insertions of any of the TE families found to be active in the somatic cells of the gut.

Altogether, with the use of long-read sequencing, we provide further evidence supporting active transposition in the normal fly intestine. Furthermore, our data comparing gut and head DNA as well as inherited *de novo* insertions indicate that 1) distinct TEs can be active in the different somatic tissues and 2) active TEs may differ between the soma and the germline.

**Somatic transposition is enriched in genes and regions of active enhancer-like chromatin**

The identification of hundreds of somatic TE insertions with base-pair resolution allowed us to address whether somatic transposition acted uniformly throughout the genome or specifically affected particular genomic regions. To achieve this, we analyzed genomic and chromatin features of the *de novo* somatic TE insertion sites.

Analyzing the genome-wide distribution of somatic TE insertions from the clonal gut samples in mappable regions of the genome revealed that integrations occurred broadly across *Drosophila* chromosomes (Fig 6A). Importantly, somatic transposition was very frequent in genic regions. Although they were depleted from coding sequences, somatic LTR element integration sites were significantly enriched in introns (*rover* and *copia* insertions) and 3'UTRs (*rover* insertions) (Fig 6B). Similar genic enrichment was evident in the singleton insertions of *rover LTR* identified by the long-read sequencing of normal gut DNA pools (Fig EV4), suggesting that it was not influenced by the clonal expansion. Importantly, in both data sets, we found insertions within or nearby (<500bp upstream of the TSS) genes with established roles in the regulation of gut and ISC homeostasis (Fig 6C, Tables EV5 and EV6). Apart from the insertions in *Notch*, which were selected for in clonal samples, we

detected insertions in genes involved in the EGFR (*pointed*, *EGFR*), JNK (*puckered*), JAK/STAT (*Stat92E*), Wnt (*frizzled*), insulin (*Insulin-like receptor*) and VEGF (*Pvf1-3*) pathways, as well as chromatin modifiers (*kismet*, *osa*) and other regulators of ISC homeostasis. Some of the affected genes were hit multiple times (Fig 6C, Tables EV5 and EV6). It is possible that these genes are hot-spots for TE insertions, perhaps due to genome sequence or structure. Alternatively, these insertions could drive positive selection of the resulting cell lineage through promoting stem cell proliferation, resulting in their post-insertion enrichment. Thus, we find that somatic TE insertions are enriched in genic regions, including those with important functions in the gut.

To further probe into a genome-wide distribution of somatic TE insertions, we investigated the overlap between candidate integration sites mapped in clonal (Fig 6D) and pooled gut samples (Fig EV4B) with publicly available *Drosophila* modENCODE datasets profiling chromatin features and transcription factor binding sites (The modENCODE Consortium *et al*, 2010). Comparing clonal gut TE insertion sites of LTR elements (*rover* and *copia*) with tracks from adult fly tissues revealed significant depletions from genomic regions enriched in silent chromatin features, such as methylated histone H3 (H3K9me2/3, H3K27me3), heterochromatin protein 1a (HP1a) or linker histone H1 (Riddle *et al*, 2011). Although less strongly, LTR element insertion sites were also depleted from regions marked by H3K36me2/3, typically associated with exons of transcribed genes (Kharchenko *et al*, 2011). This negative correlation is in agreement with the depletion of TE insertion sites from coding regions as documented above (Fig 6B). In contrast, we observed a strong positive correlation between LTR element integration sites and genomic regions rich in acetylated histone H3 (H3K18ac, H3K27ac) and H4 (H4K8ac) as well as H3K36me1, marks associated with active promoters, transcribed regions and enhancers (Kharchenko *et al*, 2011; Nègre *et al*, 2011). *De novo* insertions were also significantly enriched in genomic regions bound by LSD1/Su (var)3-3, a histone lysine-demethylase responsible for removing histone H3K4-methyl marks from active promoters (Shi *et al*, 2004; Stefano *et al*, 2007). The correlations for LTR element insertions identified in clonal and pooled DNA samples were very similar (Figs 6D and EV4B), suggesting that the distribution of somatic insertion sites was the same for normal and neoplastic tissue. Correlations of the LINE-like *I-element* insertion sites were much weaker than those obtained for LTR elements and should be interpreted with caution due to a low total number of insertions (Fig 6D). However, the enrichment in H3K36me1 was also significant for this non-LTR TE family. Consistent with the insertion timing analysis implying that transposition could act pre-clonally and during

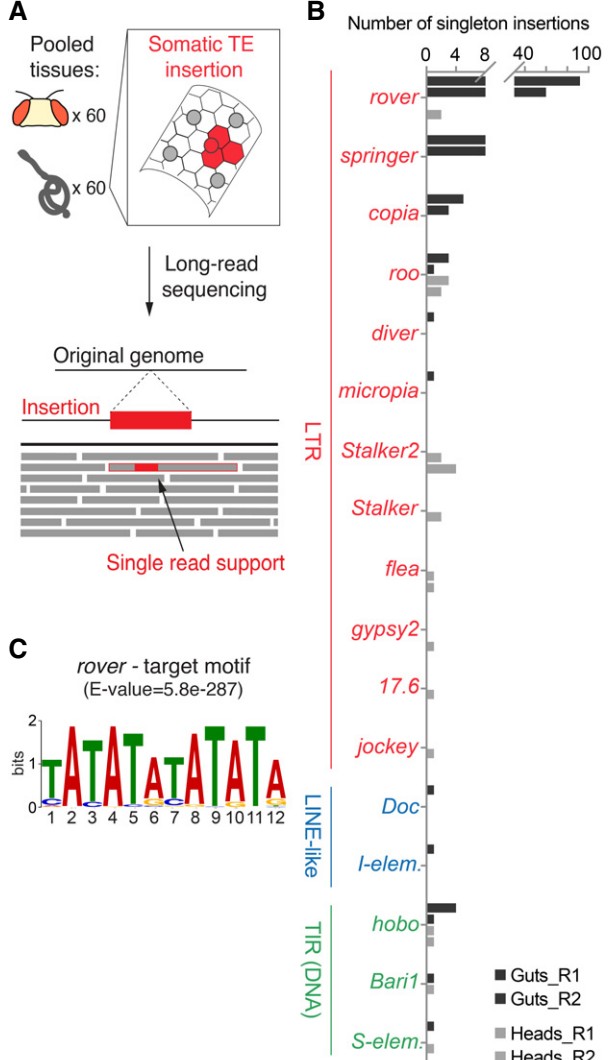

**Figure 5. Long-read sequencing implies tissue-specific transposon mobility.**

A Genomic DNA was isolated from pools of 60 guts or 60 heads dissected from the same individuals and subjected to ONT long-read sequencing. If a somatic insertion occurs in a tissue and does not undergo clonal expansion, it will be present in a small fraction of cells within the pool sequenced. Long-read sequencing allows to identify putative somatic integrations as rare TE inserts, fully contained in a single continuous sequencing read and generating a valid TSD.

B TE families with tissue-specific, singleton TSD-bearing insertions detected in pooled gut or head DNA.

C The target site motif found around (± 10bp) gut-specific *rover* singleton insertion sites identified with the long-read sequencing (E-value was calculated with MEME (Bailey *et al*, 2009), where E-value < 0.05 is considered statistically significant).

Data information: In (B), R1 and R2 are two biological replicates.
Source data are available online for this figure.

development (Fig 3B), comparable enrichments and depletions were also found for short-read clonal (Fig EV5) and long-read bulk (Fig EV4 B) gut sequencing datasets with the modENCODE tracks derived from *Drosophila* larval stages.

To confirm whether similar TE insertion site enrichment could be observed with gut-specific chromatin features, we used our recently published DamID profiles of chromatin factors in intestinal stem cells (Gervais *et al*, 2019) (Fig 6E). In agreement with the results obtained with the modENCODE datasets, somatic TE insertions from clonal gut samples were highly enriched in genomic regions bound by chromatin modifiers Kismet and H3K4 mono-methyl-transferase Trithorax-related (Trr) as well as RNA polymerase II (Pol II) (*rover*, *copia*, and *I-element*). All these factors were previously shown to map open, transcriptionally active chromatin (Marshall & Brand, 2017; Gervais *et al*, 2019). Concurrently, somatic transposition sites of LTR elements were strongly depleted from repressed chromatin domains bound by HP1 (a reader of histone H3K9me3) (Fig 6E). Finally, enrichment in chromatin domains bound by Kismet and Pol II, and depletion in HP1-bound sites were also significant for putative somatic singleton insertions identified in non-clonal gut DNA pools (Fig EV4C), suggesting that the insertion site enrichment in active chromatin also occurred in normal tissues without clonal expansion.

Taken together, our data revealed non-random distribution of retrotransposon insertions sites in a somatic tissue *in vivo*. Somatic transposition frequently affects genic regions of the genome and is enriched in open, transcriptionally active chromatin.

# Discussion

Our study provides a genome-wide view of how transposable elements mobilize in a somatic tissue. We show that endogenous retrotransposons mobilize in the fly gut and create *de novo* insertions genome-wide. Somatic insertions are enriched in genes and open chromatin, and they can occur in the *Notch* tumor suppressor, likely leading to clonal neoplasia.

### Somatic retrotransposition in the fly intestine

By whole-genome sequencing of clonally expanded gut neoplasia, we were able to detect hundreds of high-confidence retrotransposition events. Clonal expansion brings an advantage of amplifying *in vivo* any genetic variant otherwise present in a tissue with a very low frequency. A similar approach has previously been taken to demonstrate somatic transposition in human tumor samples. Indeed, retrotransposition was observed in many tumor types (Iskow *et al*, 2010; Lee *et al*, 2012; Solyom *et al*, 2012; Shukla *et al*, 2013; Helman *et al*, 2014; Tubio *et al*, 2014; Doucet-O'Hare *et al*, 2015; Ewing *et al*, 2015; Paterson *et al*, 2015; Rodić *et al*, 2015; Scott *et al*, 2016; Schauer *et al*, 2018). Most of these studies failed to detect somatic integrations in matched healthy tissues, leading to the conclusion that transposition was likely limited to the cancerous state. However, careful analysis of gastro-intestinal and esophagus tissues did suggest that active retrotransposition occurring in normal cells can undergo clonal expansion in a tumor context (Ewing *et al*, 2015; Doucet-O'Hare *et al*, 2016). Likewise, here we provide evidence that TEs are expressed and mobilize in a normal fly gut, without a prerequisite of neoplastic transformation. Studies of the mammalian brain have provided different pieces of evidence suggesting that somatic transposition can occur in the embryo, during neurogenesis as well as in mature neuronal cells (Evrony

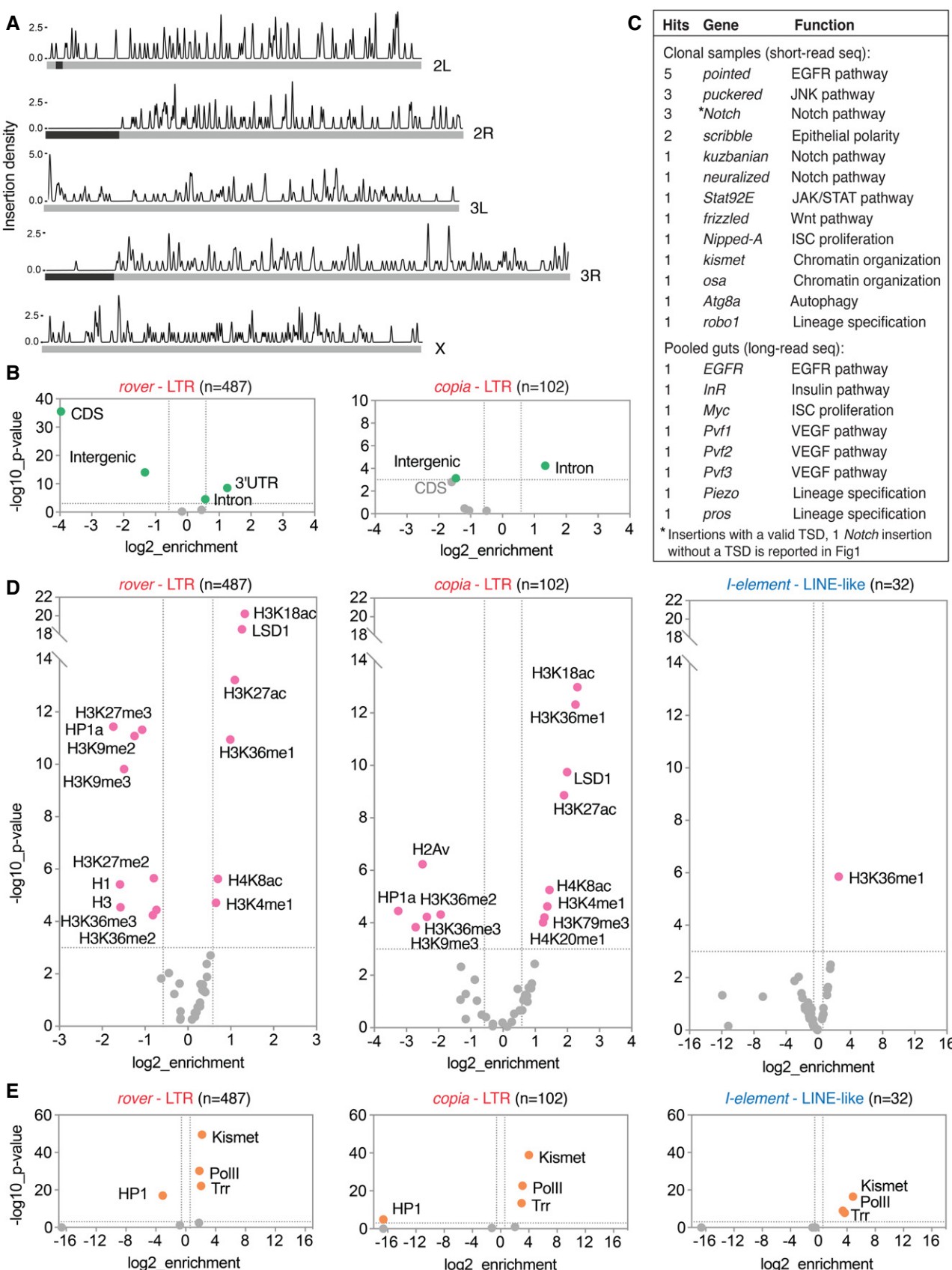

**Figure 6.**

◀ **Figure 6. TEs frequently insert in genes and regions of open enhancer-like chromatin.**

A   The distribution of somatic TE insertion sites on the *Drosophila* chromosomes. Dark gray boxes represent heterochromatic regions.
B   Somatic TE insertion sites of *rover* LTR elements were depleted from intergenic and exonic sequences and enriched in introns and 3'UTR regions of the fly genome. Insertions of *copia* LTR elements were also enriched in introns.
C   Selected genes relevant for the gut physiology with putative somatic insertion sites. Genic regions ± 500 bp were considered.
D   Correlations of somatic insertion sites of the three most represented TE families (*rover*, *copia*, and *I-element*) with modENCODE tracks for adult fly tissues.
E   Correlations of somatic insertion sites of the three most represented TE families (*rover*, *copia*, and *I-element*) with DamID tracks for adult fly intestinal stem cells (ISC).

Data information: Colored data points and labels highlight significant positive or negative correlations (Fisher's exact test with Benjamini–Hochberg correction, $P < 0.001$, $-1.5 >$ enrichment$>1.5$). *de novo* insertions from the *Pros > 2xGFP* clonal gut samples obtained with the short-read sequencing were used for all plots. Source data are available online for this figure.

*et al*, 2012, 2015; Upton *et al*, 2015; Faulkner & Garcia-Perez, 2017). Similarly, our data suggest that somatic TE insertions could also be acquired during development and gut lineage specification as well as during adult life.

### Applying long-read sequencing to detect somatic transposition

We provide further strong evidence for somatic transposition acting in a normal tissue with the use of long-read sequencing technology to assess rare TE insertions in the absence of clonal expansion. Indeed, this technology offers important benefits over classical short-read sequencing in mapping non-referenced TE insertions. It enables full-length detection of inserts within one sequencing read, resolving not only both ends but also the entire length of an insert. Moreover, it outperforms short-read technology in the analysis of low-complexity genomic regions where short-read mapping poses particular difficulty. Thus, rare somatic TE insertions can be detected from pooled DNA libraries, even if supported by a single sequencing read. However, robust controls need to be implemented to help to exclude germline variants in a population. Additionally, as somatic insertions are very rare in the sequencing of pooled DNA, the sensitivity of this approach is lower as compared to the sequencing of clonally amplified genomes, likely allowing the recovery of only a snapshot of all somatic insertions present in the sequenced tissues. The use of single individuals, rather than pooled tissues, could rule out germline variants and improve sensitivity. Thus, by the use of long-read sequencing we put forward a novel methodology for detecting somatic TE activity. A similar approach has very recently been proposed to map rare TE germline variants in *Drosophila* (Mohamed *et al*, 2020) and to perform epigenomic profiling and non-referenced TE mapping in human datasets (Ewing *et al*, 2020), further showing that long-read sequencing will certainly gain popularity in the field. As our results obtained with this technology are highly consistent with our results obtained with short-read sequencing of neoplastic clones, we believe the singleton reads detected with long-read sequencing are very likely *de novo* somatic events.

### Transposition across different tissues

Historically, focus has been on somatic transposition in neuronal tissues, where TE mobility was proposed to contribute to functional differences between individuals (Erwin *et al*, 2014). Our data suggest that TE families active in the gut are not highly mobile in the head of the same individuals, implying that the repertoire of mobile TEs might vary between different somatic tissues. Indeed, many studies have begun to uncover tissue- and cell-specific patterns of TE expression in human and model organisms (Mietz

*et al*, 1992; Faulkner *et al*, 2009; Philippe *et al*, 2016; Deininger *et al*, 2017; Pehrsson *et al*, 2019; Chung *et al*, 2019; Ansaloni *et al*, 2019; Sanchez-Luque *et al*, 2019; Treiber & Waddell, 2020). However, in most cases, the lack of data on somatic insertions hinders the direct comparison between transcriptional activity and mobility. Importantly, here we show that in the gut tissue, TE transcript levels do not parallel insertional activity. However, we cannot exclude that active TEs could be expressed in a cell-type-specific manner and that cell-type-specific TE expression patterns could correlate better with the mobility. Alternatively, mobility might happen during earlier developmental stages and coincide with higher levels of transcription occurring at that time. Nevertheless, our data suggest that caution should be taken when using transcript levels as a proxy for TE insertional activity. How additional factors, aside from those regulating RNA transcript levels, may contribute to tissue-specific somatic TE mobilization, remains to be determined.

Apart from gut versus head specificity, we also show that somatically active TEs were not detected to be mobile in germ cells. Thus, our data speak against an overall deregulation and high retrotransposon activity, as previously documented in the germline of *Drosophila* dysgenic crosses (Kidwell *et al*, 1977; Pélisson, 1981; Rubin *et al*, 1982) or in other rare genetic backgrounds with bursts of TE activity in the germline and soma (Gerasimova *et al*, 1985; Georgiev *et al*, 1990).

### TE insertions distribute non-randomly in the somatic genomes

Mapping somatic retrotransposition insertions with base-pair resolution reveal enrichments in insertion site distribution of endogenous retrotransposons *in vivo*. Indeed, gut retrotransposon insertions are found more frequently in transcriptionally active, enhancer-like chromatin, a bias that is similar to the insertion site enrichments previously observed for the murine leukemia virus (MLV) and the PiggyBac transposon in human T-cell cultures (Gogol-Döring *et al*, 2016; Sultana *et al*, 2017, 2019). Similarly, recent analysis of TE insertion sites in human cancer genomes revealed enrichment in DNase hypersensitive open chromatin and depletion in histone H3K9me3-rich heterochromatin (Rodriguez-Martin *et al*, 2020). Our data uncover similar insertion site enrichments *in vivo*, not only in the context of neoplastic clones but also in a normal tissue. In the fly gut, as transposition acts in a renewing and dividing tissue, the uncovered insertion site distribution is likely a result of pre-insertion target site choice as well as post-insertion selection in the tissue, as previously demonstrated for *de novo* L1 insertions in human culture cells (Sultana *et al*, 2019). Negative selection probably contributed to the significant depletion of TE insertions in coding regions, as such insertions, presumably deleterious, would lead to their

elimination from the tissue by clonal competition. Accordingly, insertion site enrichment in genic regulatory regions (UTRs and introns) could suggest a beneficial impact of transposition on the positive selection of cells with somatic TE insertions. Enrichment of *de novo* insertions in open chromatin could, at least partially, be explained by physical DNA accessibility. However, we cannot exclude that other, yet unknown mechanisms also act at the pre-insertion level to direct retrotransposition away from exons and silent chromatin but toward non-coding genic regions and active chromatin.

**Consequences of somatic transposition**

The impact of transposition on the biology of somatic tissues is under debate, as is its contribution to disease and aging (Faulkner & Garcia-Perez, 2017; Chuong *et al*, 2017; Dubnau, 2018). Here, we report evidence for somatic transposition with a functional impact on an adult tissue, by retrotransposon insertions into a tumor suppressor gene *Notch*. As spontaneous neoplasia are isolated based on the characteristic *Notch* loss of function phenotype, and in those samples, we found no other somatic events genome-wide that could explain inactivation of the Notch pathway (see also preprint: Riddiford *et al*, 2020), it is very likely that the somatic LTR retrotransposon insertions in *Notch* where indeed causative for neoplasia formation. Similarly, in mice somatic LTR element insertions causing oncogene or cytokine gene activation have been previously reported (Mietz *et al*, 1992; Howard *et al*, 2008). In human somatic, *de novo* L1 retrotransposition activating oncogenic pathways has been documented in colorectal cancer (Miki *et al*, 1992; Scott *et al*, 2016) and hepatocellular carcinoma (Shukla *et al*, 2013), leading to a hypothesis that individual variation in somatically active elements could represent a novel form of cancer risk (Scott *et al*, 2016).

In addition to *Notch*-inactivating events, many genome-wide retrotransposon insertions identified by us occurred in genic regions, including genes directly implicated in the tissue physiology. We uncover exonic integrations, which would most likely lead to gene inactivation, as well as insertions in intronic or UTR sequences. Numerous examples of germline transposition show that TE insertions in non-coding genic regions can affect, both positively or negatively, target gene expression (Shen *et al*, 2011; Gong & Maquat, 2011; Mateo *et al*, 2014; Ong-Abdullah *et al*, 2015; Ding *et al*, 2016; Van't Hof *et al*, 2016). As for the *Notch* gene, in male flies, an X chromosome insertion would result in a modification of a single available gene copy. In contrast, when two alleles are present (autosomes and female X chromosome), a somatic insertion would likely inactivate only one of them, limiting the functional impact. However, such insertions could still result in hypomorphic phenotypes. Moreover, unaffected alleles could be lost due to secondary genetic events of loss of heterozygosity, which we have shown to occur frequently in the fly gut (Siudeja *et al*, 2015; Siudeja & Bardin, 2017). Finally, apart from directly affecting coding regions, non-genic transposable element insertions occurring in active chromatin could contribute cis-regulatory elements acting on neighboring or even distant genes leading to gain-of-function or misexpression, as previously demonstrated for germline transposition (reviewed in Chuong *et al*, 2017). Hence, we hypothesize that transposition acting on genes or regulatory regions in the ISCs could influence

stem cell fitness. By doing so, TE mobility could influence the clonal selection in the tissue by eliminating some and favoring other stem cell genomes.

Since somatic transposition may have functional consequences and contribute to diseases or aging, understanding how somatic transposition is controlled and how tissue-specificity arises, is of keen interest. We provide evidence that *Drosophila* will be an insightful model system for addressing mechanisms of somatic TE control and physiological consequences of somatic transposition.

# Materials and Methods

## Experimental techniques

### *Drosophila* stocks and husbandry

*Pros > 2xGFP* adults were obtained by crossing *Pros$^{V1}$GAL4/ TM6BTbSb* females (J. de Navascués) with *UAS-2xGFP*; males (Bloomington). *Dl > nlsGFP* were obtained by crossing *DlGal4/ TM6TbHu* (Wang *et al*, 2014) females with *UAS-nlsGFP* males (Bloomington). Flies were maintained on a standard medium at 25°C with a day/night light cycle. For crosses, 10–15 females were mixed with males in standard vials. Progeny were collected over 2–4 days after eclosion. Adults were aged in plastic cages (mixed males and females) (10 cm diameter, 942 ml, 700–900 flies/cage) with freshly yeasted food provided in petri dishes every 2–3 days. Every 7 days, flies were transferred to clean cages.

### Tissue isolation and short-read DNA sequencing

6- to 7-week-old *Pros > 2xGFP* or *Dl > nlsGFP* males were used to isolate spontaneous neoplasia from unfixed tissues. Clones were identified by the accumulation of GFP-positive cells. The midgut region containing an estimated 30%–80% neoplastic cells (which represents > 80% of neoplastic DNA) was manually dissected and transferred immediately to a drop of the ATL Buffer (Qiagen) for subsequent DNA isolation. Neighboring control gut tissue as well as the fly head and the fly thorax were also dissected. Genomic DNA was isolated with the QIAamp DNA MicroKit (Qiagen) according to the manufacturer's protocol for processing laser-microdissected tissues. DNA quantity was measured with Qubit dsDNA High Sensitivity Assay Kit (Thermo Fisher Scientific). Genomic DNA libraries were prepared with the Nextera XT protocol (Illumina) using 0.6 ng of starting material. Whole-genome 2X100 bp or 2X150 bp paired-end sequencing was performed on HiSeq 2500 or Novaseq 6000 (Illumina). Expended View Table EV1 provides basic sequencing statistics for all samples used in this study.

### PCR validation of somatic TE insertions in Notch and Sanger sequencing

For full-length PCR validation, we designed PCR primers up- and downstream of identified candidate insertions to amplify either a short wild-type genomic DNA fragment or its variant with a TE insertion. PCR was performed on 0.5–1 ng of genomic DNA with the LongAmp Taq DNA Polymerase (New England Biolabs) using standard PCR reaction mix and long extension times (9 min at 65°C). In sample P51, full-length PCR amplification did not produce a band. Nevertheless, we successfully validated the 5' breakpoint of this insertion with one primer in the TE sequence and one in the

downstream genomic sequence. All primers used for the validation are listed in Appendix Table S1. The DNA fragments containing TE insertions were further PCR-amplified and gel-purified. The amplified products were then Sanger sequenced.

### RNA and small RNA isolation and sequencing

For RNA isolation, gut and head tissues from 1-week-old flies were dissected in cold, RNase-free PBS, transferred to 100 μl of TRIzol Reagent (Thermo Fisher Scientific), homogenized with a plastic pestle and snap-frozen in liquid nitrogen for storage at −80°C. Upon thawing, samples were further processed according to the TRIzol Reagent manufacturer's protocol. Purified RNA was treated with DNase (Ambion) for 1 h at 37°C, further purified with phenol-chloroform extraction and isopropanol precipitation, and resuspended in RNase-free water. All samples had A260/280 ratios above 1.9 and A260/230 ratios above 2.0. RNA integrity was checked on Bioanalyzer (Agilent) using the Agilent RNA 6000 Nano Kit and concentrations were assayed with the Qubit RNA Broad Range Assay Kit (Thermo Fisher Scientific). For the transcriptome analysis, 700ng of total RNA was used to prepare libraries according to the TruSeq Stranded mRNA protocol (Illumina). Samples were processed in biological triplicates. 2X100 bp paired-end sequencing was performed on Novaseq (Illumina). Small RNA fractionation and sequencing was performed by Fasteris, SA (Geneva, CHE). Briefly, after 2S rRNA depletion and PAGE gel-sizing for 18–30-nt fragment size, libraries were prepared according to the TruSeq small RNA Kit (Illumina) and sequenced on NextSeq 500, in 1 × 50bp single-end mode. Samples were processed in biological duplicates.

### High molecular weight genomic DNA isolation and long-read sequencing

For sequencing of non-neoplastic tissues, we isolated guts from 25-days-old female flies without any visible neoplasia along with the fly heads. Tissues were dissected in ice-cold, nuclease-free PBS and snap-frozen in liquid nitrogen before DNA isolation. High molecular weight genomic DNA was isolated from pools of 60 guts or 60 heads with the MagAttract HMW DNA Kit (Qiagen) according to manufacturer instructions. gDNA was eluted with nuclease-free water. DNA integrity was verified on a 0.6% agarose gel and concentrations were measured with Qubit dsDNA Broad Range Assay Kit. All samples had A260/280 ratios above 1.8 and A260/230 ratios above 2.0. Libraries were prepared with 800ng of DNA following the 1D Genomic DNA by Ligation Protocol (SQK-LSK109, Oxford Nanopore Technologies). Sequencing was performed on MinION using R9.4.1 flow cells (Oxford Nanopore Technologies) and 48 h-long sequencing runs. Expended View Table EV1 provides basic sequencing statistics for all samples used in this study.

### Computational analysis

### Putative TE insertion detection from short-read DNA sequencing

The following applies to all Illumina short-read paired-end DNA sequencing datasets. Adapter sequences were trimmed using fastp version 0.19.5 (Chen et al, 2018). Trimmed reads were aligned to release 6.13 of the Drosophila melanogaster reference genome (FlyBase) using bwa-mem version 0.7.17. bwa-mem parameters were default parameters, except for -Y (use soft clipping for supplementary alignments) and -q (don't modify mapQ of supplementary

alignments). Trimmed reads were also aligned to Drosophila melanogaster TE family consensus sequences (https://github.com/bergmanlab/transposons) using bowtie2 (Langmead et al, 2009) version 2.3.4.3. Duplicate reads were marked using picard markdup 2.18.2 (http://broadinstitute.github.io/picard/). Genome alignments and TE alignments are inputs to the readtagger command of the readtagger package (https://github.com/bardin-lab/readtagger). readtagger writes SAM tags for alignments where either the alignment or the mate of the alignment also aligns to a TE or other non-reference genome sequence. Tags contain information about the alignment (TE reference, alignment start, alignment end, query start, query end, and alignment orientation), can be visualized in IGV (as illustrated in Fig 1B), and were used to locate potential non-reference TE insertions. As reads at one insertion site can show homology to different reference TEs of the same family, they are often visualized with different colors. Next, the findcluster command takes the tagged alignment files and iteratively splits and groups tagged reads within a distance that corresponds to the 95% interval of the insert distance into clusters based on their alignment orientation and clipped sequences, and annotates if any cluster shows signs of a target site duplication (TSD). These unfiltered clusters are further linked to soft-clipped sequences at the 5' and 3' ends of putative insertions, so that the presence of a particular clipped sequence at a given genomic position can be used as a proxy for determining whether reads are congruent with an insertion or not.

The output of the findcluster step is a GFF file containing putative insertions and soft-clipped sequences and their genomic position as well as a BAM file containing only aligned reads assigned to a cluster (this includes alignments that support an insertion and alignments that support the reference).

### Characterizing non-reference germline insertions

Extended View Tables EV7 and EV8 list all detected non-reference TE insertions with a valid TSD that were classified as germline based on read evidence found in both tissues of the same individual (gut and head). Insertion sites with median coverage across all samples below 10 were discarded. For the Appendix Fig S4 "germline shared" insertions were those detected in all 34 individuals of the Pros > 2xGFP flies, and "germline private" insertions were found in only a single individual (in both gut and head).

### Filtering putative TE insertions to obtain non-reference somatic TE insertions

findcluster outputs for each sample were filtered to retain only insertions that contain both mate and split read support. These putative insertions were then processed with the confirm_insertions command of the readtagger package. This command takes as input a file containing pre-filtered putative insertions in GFF format, a set of all putative insertions from all samples and a set of insertions from all samples that can be considered a panel of normal. To detect somatic TE insertions for a particular tumor dataset, the panel of normals are all head datasets. Inversely, to detect somatic head insertions, the panel of normal are all tumor datasets. confirm_insertions links insertions from all samples using overlapping clipped sequences, genomic location, and the family of a putative TE insertion. Putative somatic TEs were those insertions that were not found within the panel of normal. We further required a valid target site duplication to be present. For each candidate somatic TE insertion,

we generated an IGV screenshot that includes 500 nucleotides up- and downstream of each insertion. Upon visual inspection of screenshots, putative insertions that were likely incorrectly called due to imprecise annotation of either the putative insertion or the control insertion were discarded. The lists of all identified putative somatic insertions can be found in Extended View Table EV2 (gut clonal samples) and EV3 (head samples).

### Filtering putative TE insertions to obtain non-reference germline insertions

To estimate the rate of germline transposition, datasets were analyzed as for somatic TE insertions, but we treated each family individually, where the panel of normal constituted all other families. The retained insertions were then private to the family being analyzed.

### Comparison of somatic TE allele frequencies to neoplasia-initiating events

Allele frequencies of somatic insertions were estimated from the total pool of read pairs that overlap a putative insertion by dividing the number of read pairs (a single read-pair can only contribute once) directly supporting an insertion by the sum of supporting plus opposing reads. To estimate the fraction of cells carrying an insertion in Fig 3B and C, we used the calculated allele frequencies of sex chromosome TE insertions and we adjusted the allele frequencies of TE insertions on autosomes by multiplying with a factor of 2. Allele frequencies of neoplasia-initiating events were taken from a companion paper addressing structural variation in the same model system(preprint: Riddiford et al, 2020). As for TE insertions, the number of reads directly supporting each event was divided by a number of supporting + opposing reads. In complex variants with multiple breakpoints, highest allele frequency for the variant was taken. The exception to this were samples P15, P47, and D5, where somatic TE insertions in Notch reported in this study were assigned as neoplasia-initiating events. For sample P15, with two somatic insertion in Notch, the insertion with a higher allele frequency was set as the putative Notch-inactivating event. Estimated cell fractions of tumor-initiating events below 1 indicate that sequenced samples contained the expanded clone and the adjacent normal tissue. We plotted the estimated percent of cells with an insertion as dots and the tumor-initiating event as bars.

### Long-read sequencing data analysis

Nanopore reads were basecalled using guppy version 3.2.4. Since read length and sequencing depth is not uniform for long-read datasets and this can affect the number of full-length TE insertions that are detectable, reads were normalized using the normalize_readsizes of the readtagger package. All analyzed nanopore libraries therefore have the same read-size distribution and sequencing depth.

Reads were aligned to release 6.13 of the Drosophila melanogaster reference genome using minimap2 version 2.17 (Li, 2018) with the -Hk19 preset for Nanopore reads and the -Y flag. Alignments with a mapping quality below 40 were discarded with samtools view. extract_variants from the readtagger package was used to check all soft-clipped or insert sequences for homology to TEs using mappy. Aligned positions around soft-clipped or insert sequences were written to a new alignment file along with a tag that describes the TE alignment. For soft-clipped sequences, a single-aligned N-nucleotide was written out together with the soft-clipped sequence. For inserts,

the insert sequence was written out using 1 flanking N-nucleotide at each site. Alignment files were then parsed into a tabular format and analyzed to find unique full-length transposable element insertions using ipython notebooks available at https://github.com/bardin-lab/somatic-transposition-fly-intestine. The lists of all identified singleton insertions can be found in Extended View Table EV4.

### Motif analysis at integration sites

Motifs were determined by extracting 10 flanking nucleotides upstream and downstream of each insertion using bedtools slopbed and bedtools getfastabed (Quinlan & Hall, 2010) and running meme version 5.0.5 (Bailey et al, 2009) on the resulting multi-fasta file. Meme parameters were -dna for DNA alphabet, -revcomp which checks the reverse complement for motifs, -pal for checking for palindromes and a motif width between 9 and 50 (-minw 8, maxw 50).

### Genome features enrichment analysis

Pre-analyzed modencode datasets (The modENCODE Consortium et al, 2010) were downloaded and lifted over to release 6.13 of the Drosophila genome. DamID peaks (Gervais et al, 2019) were from GSE128941. Overlap was analyzed in ipython notebooks available at https://github.com/bardin-lab/somatic-transposition-fly-intestine. pybedtools fisher (Dale et al, 2011) was used to determine enrichment and significance of overlap. P-values were adjusted with Benjamini–Hochberg correction (alpha = 0.05). Correlations with P-value < 0.001 and $-1.5 >$ enrichment $> 1.5$ were considered significant. The overlap between genes and TE insertion sites was calculated with bedtools windowbed using $\pm$ 500 bp window size.

### RNA-seq data analysis

For RNA-seq analysis, reads were trimmed off their adapters using Atropos (Didion et al, 2017) and quasi-mapped against the Drosophila reference transcriptome (release 6.13, Flybase) supplemented with family-level TE sequences (https://github.com/bergmanlab/transposons) using Salmon version 0.14.1 (Patro et al, 2017) and RPM values were reported in Fig 4. Differential expression analysis (Appendix Fig S3) was performed on previously published datasets (Patel et al, 2015) using DESeq2 (Love et al, 2014). TE transcripts with an adjusted P-value < 0.01 were considered differentially expressed. We used the plot_coverage command of the readtagger package to create coverage plots.

### Small RNA analysis

Sequencing adapters were trimmed using fastp version 0.19.5. Reads were aligned to family-level TE sequences using HISAT2 (Kim et al, 2019) version 2.1.0. Size distributions were plotted using ipython notebooks. Ping-pong signatures were calculated using the Small RNA Signatures tool, version 3.1.0 (Antoniewski, 2014).

## Data availability

Datasets used for this study are available under the following accession numbers:

- whole-genome neoplasia/head control sequencing (samples P1-P6; (Siudeja et al, 2015)): ArrayExpress E-MTAB-3917 (https://www.ebi.ac.uk/arrayexpress/experiments/E-MTAB-3917/)

- whole-genome neoplasia/head control sequencing (samples P7-P66 and D1-D8): NCBI, Bioproject PRJNA641572 (https://www.ncbi.nlm.nih.gov/bioproject/?term = PRJNA641572)
- Nanopore ONT sequencing of pooled guts/heads; RNA-seq of pooled guts; small RNA-seq of pooled guts and ovaries; Pros > 2xGFP parents/offspring DNAseq: EBI, study accession number PRJEB41757 (https://www.ebi.ac.uk/ena/browser/view/PRJEB41757)

**Expanded View** for this article is available online.

## Acknowledgements

High-throughput sequencing has been performed by the ICGex NGS platform of the Institut Curie supported by the grants ANR-10-EQPX-03 (Equipex) and ANR-10-INBS-09-08 (France Génomique Consortium) from the Agence Nationale de la Recherche ("Investissements d'Avenir" program), by the Canceropole Ile-de-France and by the SiRIC-Curie program—SiRIC Grant "INCa-DGOS-4654." We thank C. Antoniewski, P. Hollyoak, T. Hall, and W. Hamitou for their contribution to some experiments and data analysis; N. Da Cruz and B. Leveille Nizerolle for technical assistance; as well as members of the Bardin team, N. Servant, J. Waterfall, D. Bourc'his, M. Greenberg and P.-A. Defossez for discussions and comments on the manuscript. This work was supported by Fondation pour la Recherche Médicale (A. B., DEQ20160334928), as well as funding from the program "Investissements d'Avenir" launched by the French Government and implemented by ANR, ANR SoMuSeq-STEM (A. B), Labex DEEP (ANR-11-LBX-0044), IDEX PSL (ANR-10-IDEX-0001-02 PSL), and ICGex STEM-SOM-GERM (AJB and KS). Salary support of KS is from Inserm; AJB from CNRS; NR is through a grant from Fondation ARC (PDF20161205270), and BB by ENS Lyon.

## Author contributions

KS, MB, and AJB designed the study and analyzed the data. MB developed software and performed all bioinformatic analysis. KS performed a majority of the experiments. BB collected samples for the Delta > nlsGFP background; MS obtained data for Fig EV3; and AW contributed to the long-read sequencing experiments. NR provided variant frequencies of neoplasia-initiating events for Fig 3B. SL prepared samples for sequencing. KS and AJB wrote the manuscript with contribution from MB and other authors.

## Conflict of interest

The authors declare that they have no conflict of interest.

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
