## [Review Process File · The EMBO Journal]

Unravelling the features of somatic transposition in the *Drosophila* intestine

Katarzyna Siudeja, Marius van den Beek, Nick Riddiford, Benjamin Boumard, Annabelle Wurmser, Marine Stefanutti, Sonia Lameiras, Allison J. Bardin

DOI: [10.15252/embj.2020106388](https://doi.org/10.15252/embj.2020106388)

Corresponding authors: Allison Bardin (allison.bardin@curie.fr), Katarzyna Siudeja (Katarzyna.Siudeja@curie.fr)

Review Timeline:

Submission Date:	30th Jul 20
Editorial Decision:	3rd Sep 20
Revision Received:	8th Dec 20
Editorial Decision:	11th Jan 21
Revision Received:	20th Jan 21
Accepted:	27th Jan 21

Editor: Ieva Gailite

Transaction Report:

Thank you for submitting your manuscript for consideration by The EMBO Journal. I apologise for the protracted review process of your manuscript caused by delays in review submission. We have now received two referee reports on your manuscript, which are included below for your information.

As you will see from the comments, both reviewers appreciate the novelty and the quality of the study, while pointing out several aspects that would have to be improved in the revised manuscript in order to provide more detailed insights into the dataset and to clarify the approach taken in data acquisition and analysis. Based these very positive assessments, I would like to invite you to address the concerns raised by the reviewers in a revised manuscript.

We have extended our 'scooping protection policy' beyond the usual 3-month revision timeline to cover the period required for a full revision to address the essential experimental issues. This means that competing manuscripts published during revision period will not negatively impact on our assessment of the conceptual advance presented by your study. Please contact me if you see a paper with related content published elsewhere to discuss the appropriate course of action. I would also be happy to discuss the revision in more detail via email or phone/videoconferencing.

Please feel free to contact me if you have any further questions regarding the revision. Thank you for the opportunity to consider your work for publication. I look forward to receiving the revised manuscript.

Referee #1:

This is an outstanding manuscript that addresses an important question in biology, and it makes a significant contribution. While historically, most studies of transposable elements have focused on germline, where new insertions are adaptive for the mobile element and can directly impact genetic diversity across populations, there is a growing awareness over the past decade that somatic transposition events also can take place. Such somatic events have been seen in the context of normal development, for example in brain. And also in the context of cancer, aging, and neurodegenerative disease. So there is a growing awareness that this is a fairly important but understudied topic.

The study of somatic transposition is incredibly challenging. Such events have been detected using engineered reporters from individual elements. This offers the advantage of allowing an imaging approach, and allowing structure function studies. But such approaches do not permit one to have a genomic view of which elements are active in which cells or to ask where their new inserts tend to go. DNaseq approaches solve these problems in principle. But the fact that many events are 'private' ones that are unique to a single cell or a few cells cause signal to noise issues that are so severe that they are in the range of the noise that comes from technical artifacts. So such studies remain controversial.

This study takes a really elegant approach to solve some of these problems. The authors use the clonal expansion that is seen with neoplasia in the fly gut to increase the signal to noise. They detect rampant mutations, including those caused by de novo insertions of transposons.

Importantly, they show convincing evidence that some of these insertions, in the notch gene, are causal of the neoplasia. And they show that secondary insertions also occur after the clonal expansion begins.

Another technical advance shown, which I greatly appreciate, is the addition of long read seq. This allows for detection of rare events, and sort of solves the issue of validation because it allows for full length seq through the element including the junctions.

There are plenty of novel findings here. In addition to being one of the best datasets that I have seen for somatic transposition sequencing, there are insights into the target site distribution, the cell type and tissue specificity, the diversity of elements that are mobile, the (lack of) correlation with expression level (RNAseq), the (lack of) correlation between expression and siRNA expression, the (near absence of) involvement of piRNAs, and the strong correlations with chromatin marks. This is a wealth of data that will prod the field forward significantly.

So I am enthusiastic.

I have few critiques of the data or analyses or text. The text is quite well written, and is presented clearly.

I have some suggestions for the text that I offer to the authors if it is helpful to improve the manuscript, and to place their work in a context that I think was under-emphasized.

Throughout the ms I was wondering how insertions in Notch could cause clonal expansion since N is a tumor suppressor gene. I had forgotten that N is on the X chromosome, so I kept wondering if the authors were suggesting that there were two hits that inactivated both alleles. In the discussion, the authors hit on this. They are correct that this could explain the effect in males. But in females, there would either need to be two hits or a dominant impact. Either of these are possible, but I think this warrants some discussion. Given the frequencies, is it possible that there are two hits? one could be transposon driven and the other by some other means. Second, I wondered about lateral inhibition and dosage sensitivity of N. I admit to being a neuroscientist who doesn't know the mechanistic details of Notch involvement in the gut. But I do remember the classic work on lateral inhibition in which minor fluctuations in N and D levels get amplified. In the now classic work of Pat Simpson from the 1990s or so, even a 1X vs 2X copy number change in N in adjacent cells can have a huge impact. Is this at play here?

Second, although I loved the singlets part of the story, I wonder if is worth inserting a caveat or two. The authors make a convincing argument that the profile of events seen with singlets from pooled tissue and long read is so similar to that seen with the clonal analysis that it smells real. On the other hand, these are singlets. Maybe one sentence or word of caution is warranted?

Finally, I felt that throughout the intro and discussion, the authors sort of lumped aging and neurodegeneration and normal brain transposition into one pot. It is of course true that neurodegeneration is a disease of aging. It also is true that somatic events in normal brain is where a lot of this started. But these are different things. In my view, the work on normal developmental transposition in mammalian and fly brain is a fascinating phenomenology searching for a biological impact. But transposition during aging and during neurodegeneration is a field with some real mechanistic meat and some causal implications. I think that area should be discussed a bit more. Full disclosure: I study transposons and neurodegeneration in flies and so I admit the possibility that I'm tooting my horn a bit here. so accept if you feel this critique is warranted and reject if you think its not. I won't hold that against you.

Overall, I am quite enthusiastic and really enjoyed reading this manuscript.

Josh Dubnau

Referee #2:

In this manuscript, the authors nicely demonstrate that somatic retrotransposition contributes to genomic plasticity in the *Drosophila* gut. Previous reports have shown that retrotransposons can be active in the human oropharyngeal and intestinal systems, in normal and cancerous tissues, but this is the first evidence showing this process in a model experimental system such as *Drosophila*. This approach allow them to address a very important question related to somatic mutations, especially in cancer : do they occur prior to clonal expansion and just become detectable as a result of cell growth, do they cause clonal expansion, or do they occur throughout clonal expansion as a side effect of increased genome instability. Finally, they also show that different sets of retrotransposon families are active in different somatic tissues, comparing intestine and brain. Altogether, this manuscript represents an important contribution to the fields of genomics, stem cells and aging and will be of very high interest to readers of the EMBO Journal. However, I recommend some revisions as outlined below to clarify some aspects of the presented results:

1. The performance of the sequencing and computational approaches are difficult to appreciate from the presented data. In particular, the reporting of inherited/germline insertions, which could help benchmarking the experimental process, is lacking : what is the coverage for each of these insertions, the number of supporting split reads or mates ? what is the distribution of the coverage (homogenous ? highly diverse ?) ? is it similar to somatic insertions ? Assuming that the *Drosophila* strains sequenced are pure, how reproducible is the profile of inherited insertions among individuals? what is their percentage of recovery for homozygous and heterozygous insertions?
2. In addition to the above points, supplemental tables should include the description of all transposable insertions found in the genome of the sequenced samples, and not only the somatic insertions, sample by sample.
3. To evaluate the timing of retrotransposition relative to clonal expansion, the authors use the apparent allele frequency in the sequenced sample (Fig. 3).
 - a. However, how this was calculated must be clarified. A somatic insertion being necessarily heterozygote, its allele frequency in autosomal chromosomes cannot be superior to 0.5, unless loss-of-heterozygosity occurred. Thus, read frequency should be divided by 2 rather than multiplied by 2 as indicated in Fig. 3B. Most likely, I presume that the authors have actually reported the % of cells containing a somatic insertion. Whether there is a mistake in the calculation of the allele frequency, or just an inaccurate wording, this should be clarified by detailing the precise formula used and reporting the values as supplemental data. Indeed, this could significantly impact the conclusions relative to the timing of retrotransposition.
 - b. The green horizontal line named 'onset of neoplasia' represent the 'allele frequency' of Notch-inactivating event. How this was calculated should be clarified, too. I assume that the sequenced samples contain both the expanded clone and adjacent normal tissue, otherwise the allele frequency of the Notch-inactivating event would be 1.0 for all samples.
 - c. I wonder what is the confidence interval for the allele frequency analysis: in other words, how much the variability of sequencing coverage at a given locus can influence allele frequency calculation. This could be estimated by looking at the variability of estimated alleles frequencies for inherited transposons.
4. With regards to enrichment analysis, each family of transposon is likely to behave differently in terms of integration preference and selective pressure. Therefore, performing the analysis separately for each (abundant enough) family may reveal more insightful than pooling all insertions

from all families.

5. Expression is a pre-requisite for retrotransposition. Thus, the absence of expression for some retrotransposon families that exhibit high numbers of de novo insertions can only be explained by expression in a very restricted subset of cells of the intestine (discussed), or by the fact that transposition occurred much earlier during the development (not discussed). Please add a caveat in the discussion. Also, if this is a matter of timing, do families with high somatic copy number and low expression (such as rover) more frequently exhibit high allele frequency suggestive of transposition in earlier stage of development, and vice-versa for families with high expression (such as copia)?

Minor points:

6. Please, add the orientation of transposons in the supplemental tables to provide a more complete description of the discovered insertions.

7. The enrichment of insertions in given regions of the genome is driven by the preference of the integration complexes for these regions, but also by natural selection, through cell death or cell growth, both for expanded clones and for bulk tissues. This is clearly discussed at the end of the manuscript, however some wording throughout the main text remains ambiguous and implicitly suggests that the observed biased reflects a biased integration process, e.g. 'preferential landing sites' (p.12), 'to further probe into integration preferences', etc. Please rephrase.

8. In samples where transposons are suspected to be causative for Notch inactivation, RNA-seq has not been performed - it is clearly challenging since it would have required to extract RNA from the same sample. Thus, it is unclear how the insertions actually impair Notch expression, and if they do. In principle, it is possible that other somatic mutations outside of Notch or in non-coding regions could phenocopy Notch inactivation. Please add a caveat and rephrase sentences such as 'we uncovered TE insertions causative for Notch inactivation' (p. 8).

9. The fact that Notch gene is carried on the X-chromosome is only mentioned in the discussion. Please make it explicit in the introduction and results since it is essential to understand that a single mutational hit in males is sufficient to drive a loss-of-function.

10. Showing reads as pairs in IGV screenshots (Fig. 1 and S1) would show more explicitly the reads supporting the empty allele. For the singleton mates, is there a meaning for the different colors?

11. Fig. 1C-D, why only rover insertions were validated, and only a single one Sanger sequenced? This is important to support the sequencing approach.

12. Fig. 2, it would be of interest to know the distribution of insertion per family and sample: are the active transposons mobilized in all samples equally or highly active in some samples, but inactive in others?

13. Fig. 2E and 5C, where is (are) the break point(s) in the target motifs?

14. In Fig.6 and similar figures, were the p-values adjusted for multiple comparisons? (some look very high)

15. It is assumed that the differences between Pros>2xGFP and DI>nlsGFP strains are strain-specific, but could they also reflect tissue-specificities? Please clarify.

Response to Referees

Referee #1:

This is an outstanding manuscript that addresses an important question in biology, and it makes a significant contribution. While historically, most studies of transposable elements have focused on germline, where new insertions are adaptive for the mobile element and can directly impact genetic diversity across populations, there is a growing awareness over the past decade that somatic transposition events also can take place. Such somatic events have been seen in the context of normal development, for example in brain. And also in the context of cancer, aging, and neurodegenerative disease. So there is a growing awareness that this is a fairly important but understudied topic.

The study of somatic transposition is incredibly challenging. Such events have been detected using engineered reporters from individual elements. This offers the advantage of allowing an imaging approach, and allowing structure function studies. But such approaches do not permit one to have a genomic view of which elements are active in which cells or to ask where their new inserts tend to go. DNaseq approaches solve these problems in principle. But the fact that many events are 'private' ones that are unique to a single cell or a few cells cause signal to noise issues that are so severe that they are in the range of the noise that comes from technical artifacts. So such studies remain controversial.

This study takes a really elegant approach to solve some of these problems. The authors use the clonal expansion that is seen with neoplasia in the fly gut to increase the signal to noise. They detect rampant mutations, including those caused by de novo insertions of transposons. Importantly, they show convincing evidence that some of these insertions, in the notch gene, are causal of the neoplasia. And they show that secondary insertions also occur after the clonal expansion begins.

Another technical advance shown, which I greatly appreciate, is the addition of long read seq. This allows for detection of rare events, and sort of solves the issue of validation because it allows for full length seq through the element including the junctions.

There are plenty of novel findings here. In addition to being one of the best datasets that I have seen for somatic transposition sequencing, there are insights into the target site distribution, the cell type and tissue specificity, the diversity of elements that are mobile, the (lack of) correlation with expression level (RNAseq), the (lack of) correlation between expression and siRNA expression, the (near absence of) involvement of piRNAs, and the strong correlations with chromatin marks. This is a wealth of data that will prod the field forward significantly.

We would like to thank the referee for his appreciation of our work and the novelty of our approaches and results.

So I am enthusiastic.

I have few critiques of the data or analyses or text. The text is quite well written, and is presented clearly.

I have some suggestions for the text that I offer to the authors if it is helpful to improve the manuscript, and to place their work in a context that I think was under-emphasized.

We have now incorporated changes in the text and Fig 1, which we detail below.

Throughout the ms I was wondering how insertions in Notch could cause clonal expansion since N is a tumor suppressor gene. I had forgotten that N is on the X chromosome, so I kept wondering if the authors were suggesting that there were two hits that inactivated both alleles. In the discussion, the authors hit on this. They are correct that this could explain the effect in males. But in females, there would either need to be two hits or a dominant impact. Either of these are possible, but I think this warrants some discussion. Given the frequencies, is it possible that there are two hits? one could be transposon driven and the other by some other means.

We agree with the referee that our explanation of the spontaneous *Notch* loss-of-function neoplasia in the results section was not clear enough. Moreover, this point was also raised by the referee #2. To clarify this we have now:

- Added the following explanation at the beginning of the results section:

(p.4) *“Since Notch is located on the X-chromosome and as such is present in a single copy in males, a single hit can lead to its inactivation (Fig 1A). In contrast, females, harboring two copies of Notch, do not or very rarely develop similar spontaneous Notch inactivation events.”*

- Added to Fig1A a schematic of *Notch* localized on a single X chromosome in XY males and its loss in neoplastic clones.

Thus, as clarified above, all neoplastic clones sequenced in this study were isolated from males. The referee, raises a valid question if similar spontaneous *Notch* loss-of-function clones can arise in females. We have previously shown that such clones were not detected in wild-type females (n= 290 guts, Siudeja et al, 2015). However, we can add here that by analyzing unpublished whole genome sequencing datasets of female clones from a different genetic background generated in our lab but unrelated to this study, we have uncovered two cases of bi-allelic genetic *Notch* inactivation. This suggest that such events are indeed possible, though are much rarer than the spontaneous events in males.

Second, I wondered about lateral inhibition and dosage sensitivity of N. I admit to being a neuroscientist who doesn't know the mechanistic details of Notch involvement in the gut. But I do remember the classic work on lateral inhibition in which minor fluctuations in N and D levels get amplified. In the now classic work of Pat Simpson from the 1990s or so, even a 1X vs 2X copy number change in N in adjacent cells can have a huge impact. Is this at play here?

This is an interesting point, however, as clarified above, we are only sequencing *Notch* loss-of-function clones from male flies, where it goes from 1 to 0 copies of *Notch*. In heterozygous *Notch*^{+/-} females, no obvious phenotypes in the gut that could suggest haploinsufficiency can be observed, unlike the wing margin, which has a haploinsufficient wing margin phenotype.

Second, although I loved the singlets part of the story, I wonder if it is worth inserting a caveat or two. The authors make a convincing argument that the profile of events seen with singlets from pooled tissue and long read is so similar to that seen with the clonal analysis that it smells real. On the other hand, these are singlets. Maybe one sentence or word of caution is warranted?

We thank the referee for his recognition of our approach to use ONT technology to map putative somatic TE insertions. Indeed, as suggested there are caveats to this approach. We have now stated this more clearly in the text:

(p.10-11) *"A potential drawback of such approach could be that a germline population variant present in a single individual could be mistaken for a somatic variant. To help to exclude such variants, we eliminated all insertions detected in both gut and head DNA pools. Additionally, we used our short-read clonal datasets to estimate that germline TE variants were rare in the Pros>2XGFP background and did not belong the same TE families that somatically active TEs. (Table EV7 and Appendix Fig S4)."*

(p.14) *"Thus, rare somatic TE insertions can be detected from pooled DNA libraries, even if supported by a single sequencing read. However, robust controls need to be implemented to help to exclude germline variants in a population. Additionally, as somatic insertions are very rare in the sequencing of pooled DNA, the sensitivity of this approach is lower as compared to the sequencing of clonally amplified genomes, likely allowing the recovery of only a snapshot of all somatic insertions present in the sequenced tissues. The use of single individuals, rather than pooled tissues, could rule out germline variants and improve sensitivity."*

Finally, I felt that throughout the intro and discussion, the authors sort of lumped aging and neurodegeneration and normal brain transposition into one pot. It is of course true that neurodegeneration is a disease of aging. It also is true that somatic events in normal brain is where a lot of this started. But these are different things. In my view, the work on normal developmental transposition in mammalian and fly brain is a fascinating phenomenology searching for a biological impact. But transposition during aging and during neurodegeneration is a field with some real mechanistic meat and some causal implications. I think that area should be discussed a bit more. Full disclosure: I study transposons and neurodegeneration in flies and so I admit the possibility that I'm tooting my horn a bit here. so accept if you feel this critique is warranted and reject if you think its not. I won't hold that against you.

Indeed, in our introduction we did not discuss important work from many labs showing the links between TEs and aging and neurodegeneration due to space limitations. Overall, we tried to limit ourselves to citing the literature providing evidence for somatic TE mobility. However, we agree with the reviewer that these aspects should be mentioned and they put our study in a broader context. We have therefore now also mentioned the mobility unrelated consequences of TE deregulation and sent the reader to more comprehensive reviews on this subject.

We have now added the following text to the introduction:

(p. 2) “Interestingly, increased TE expression in many organisms has been linked to normal tissue aging as well as pathologic conditions of neurodegeneration. Evidence suggests that TE transcription may be linked to disease pathology, however, it remains unknown to which extent TE insertional activity contributes to these phenotypes (Dubnau, 2018; Burns, 2020). Nevertheless, the gypsy retrotransposon reporter activity was shown to increase in aging *Drosophila* brain, fat body and gut (Li et al, 2013; Jones et al, 2016; Wood et al, 2016; Sousa-Victor et al, 2017; Chang et al, 2019), correlating in some cases with increased DNA damage, and suggesting that TE insertional activity could indeed play a role in age-related deterioration of somatic tissues.”

Overall, I am quite enthusiastic and really enjoyed reading this manuscript.
Josh Dubnau

Referee #2:

In this manuscript, the authors nicely demonstrate that somatic retrotransposition contributes to genomic plasticity in the *Drosophila* gut. Previous reports have shown that retrotransposons can be active in the human oropharyngeal and intestinal systems, in normal and cancerous tissues, but this is the first evidence showing this process in a model experimental system such as *Drosophila*. This approach allow them to address a very important question related to somatic mutations, especially in cancer : do they occur prior to clonal expansion and just become detectable as a result of cell growth, do they cause clonal expansion, or do they occur throughout clonal expansion as a side effect of increased genome instability. Finally, they also show that different sets of retrotransposon families are active in different somatic tissues, comparing intestine and brain. Altogether, this manuscript represents an important contribution to the fields of genomics, stem cells and aging and will be of very high interest to readers of the EMBO Journal. However, I recommend some revisions as outlined below to clarify some aspects of the presented results:

We would like to thank the referee form her/his positive feedback on our work and constructive comments, that we address below.

1. The performance of the sequencing and computational approaches are difficult to appreciate from the presented data. In particular, the reporting of inherited/germline insertions, which could help benchmarking the experimental process, is lacking : what is the coverage for each of these insertions, the number of supporting split reads or mates ? what is the distribution of the coverage (homogenous ? highly diverse ?) ? is it similar to somatic insertions ? Assuming that the *Drosophila* strains sequenced are pure, how reproducible is the profile of inherited insertions among individuals? what is their percentage of recovery for homozygous and heterozygous insertions?

We thank the referee for this suggestion. We have now also mapped all non-reference germline TE insertions in clonal samples and reported them in additional data tables (**new Tables EV7 and EV8**). For each of those insertions we report their germline/somatic status, overlap of samples in which an insertion was detected (in the case of germline), the coverage at the insertion site and the number of supporting and opposing reads for each sample. The tables report all insertions recovered from both genotypes used for short read sequencing,

however for the conclusions listed below we base ourselves only on the *Pros>2xGFP* background where many more samples were sequenced.

- First, there are no homozygous insertions on the autosomes that differ from the reference genome.
- We recovered a total of 399 non-reference, germline insertions present in all 68 *Pros>2xGFP* samples sequenced, likely representing fixed germline variants (see also **new Appendix Fig S4A**).
- “Germline private” insertions, which occur in only a single individual (both in gut and head) were also recovered and they represent an upper-bound estimate for active germline transposition in the *Pros>2xGFP* background. We found 51 germline private insertions, or on average 1.5 insertion/individual. The most abundant “germline private” insertions are of *hobo* (34 inserts), *roo* (14) and *FB* (10) elements (see table below). We recovered no “germline private” *rover* insertions and only 3 *copia* and 1 *I-element* insertion, suggesting that TEs identified to be active in the soma rarely contribute germline variants. We now also use this analysis to further support the likely somatic nature of “singleton” insertions obtained with the long-read ONT sequencing of pooled tissues (see also our response to Referee 1, **new Appendix Fig S4C**).

- To estimate the percentage of recovery of germline insertions we compared insertions between the head and gut samples and asked what was the recovery rate of germline head insertions in the respective gut samples. Doing this we find that we recover on average 88.4% (minimum 83%, maximum 92.5%) of all head insertions in the gut of the same individual.

For the clonal gut samples, this recovery rate could mean that we miss on average 11.6 % of somatic insertions per sample (false negatives). However, if on the other hand, we have an 88.4 % recovery rate head samples, we would 11.6 % chance of missing an insertion in the head that we have called in the gut and thereby falsely classifying the insertion as somatic when it actually is a germline insertion. However, by using a "panel of normal" set of controls we should avoid this problem. Indeed, in our approach to determine whether an insertion is somatic, we require that the insertion not be present in any of the control tissues (this is known as a "panel of normals" or "PON"). If, for example, a candidate insertion in the gut is present also in the head of another individual we discard it. Thus, with each additional control sample we have another 88.4 % chance of discarding a false positive somatic insertion. Hence, this greatly reduces the false positive rates in our somatic calls.

- Finally, as the referee suggested, we analyzed carefully the coverage and the number of supporting reads for germline insertions. This analysis and its comparison to the somatic insertions is detailed in our response to question 3a below (concerning variant allele frequency estimations).

2. In addition to the above points, supplemental tables should include the description of all transposable insertions found in the genome of the sequenced samples, and not only the somatic insertions, sample by sample.

As described above, we have now addressed this point by including as supplementary files tables describing germline TE insertions found in the *Pros>2xGFP* (new Table EV7) and the *Delta>nlsGFP* (new Table EV8) short-read samples.

3. To evaluate the timing of retrotransposition relative to clonal expansion, the authors use the apparent allele frequency in the sequenced sample (Fig. 3).

We have now addressed all the referee's points below, regarding the variant allele frequency analysis. Importantly, this analysis supports our general conclusion that TE mobility can occur pre-clonally in a normal gut (including causing *Notch* inactivation), as well as post-Notch inactivation in neoplastic clones. Calculated variant allele frequencies are indeed always subject to error, and, as such, they are only used as rough approximations of the transposition timing. We have stressed this in the text of the Results section:

(p. 7) *"Assuming the observed allele frequency represents the true allele frequency in the cell population, it can be used as an estimate of transposition timing."*

a. However, how this was calculated must be clarified. A somatic insertion being necessarily heterozygote, its allele frequency in autosomal chromosomes cannot be superior to 0.5, unless loss-of-heterozygosity occurred. Thus, read frequency should be divided by 2 rather than multiplied by 2 as indicated in Fig. 3B. Most likely, I presume that the authors have actually reported the % of cells containing a somatic insertion. Whether there is a mistake in the calculation of the allele frequency, or just an inaccurate wording, **this should be clarified by detailing the precise formula used and reporting the values as supplemental data.** Indeed, this could significantly impact the conclusions relative to the timing of retrotransposition.

We thank the referee for pointing out this inaccuracy. We have now provided a better clarification of how allele frequencies of somatic insertions were calculated to address these questions. Indeed, by multiplying estimated variant frequencies of autosomal insertions by 2 for the graph in figure 3B, we actually aim to reflect the fraction of cells containing a somatic insertion. The y-axis title of the graph has now been changed to “Estimated cell fraction”.

The following explanation was added to the Methods section:

(p. 20) *“Allele frequencies of somatic insertions were estimated from the total pool of read pairs that overlap a putative insertion by dividing the number of read-pairs (a single read-pair can only contribute once) directly supporting an insertion by the sum of supporting plus opposing reads. To estimate the fraction of cells carrying an insertion in Fig. 3B, we used the calculated allele frequencies of sex chromosome TE insertions and we adjusted the allele frequencies of TE insertions on autosomes by multiplying with a factor of 2.”*

We are now also reporting the calculated values of estimated variant allele frequencies of all somatic insertions in the new **Source Data File for Fig 3**.

b. The green horizontal line named 'onset of neoplasia' represent the 'allele frequency' of Notch-inactivating event. How this was calculated should be clarified, too. I assume that the sequenced samples contain both the expanded clone and adjacent normal tissue, otherwise the allele frequency of the Notch-inactivating event would be 1.0 for all samples.

Yes, the reviewer is correct. As explained in the text (p. 7), the sequenced samples contain clonal cells as well as “contaminating” adjacent normal tissue:

“A TE insertion could arise before the onset of neoplasia, either during development or in the young adult gut, and be present in some cells of the normal tissue. Upon the inactivation of Notch, a stem cell would initiate clonal expansion and, at the time of analysis, the insertion would be present in all clonal cells as well as neighboring “normal” cells isolated for sequencing along with the clone.”

We have now also provided additional explanation on the estimates of variant allele frequencies of *Notch* inactivating events in the Methods section:

(p.20-21) *“Allele frequencies of neoplasia-initiating events were taken from a companion paper addressing structural variation in the same model system (Riddiford, et al, bioRxiv 10.1101/2020.07.20.188979). As for TE insertion, a number of reads directly supporting each event was divided by a number of supporting + opposing reads. In complex variants with multiple breakpoints, highest VAF for the variant was taken. The exception to this were samples P15, P47 and D5, where somatic TE insertions in Notch reported in this study were assigned as neoplasia initiating events. For sample P15, with two somatic insertion in Notch, the insertion with a higher allele frequency was set as the putative Notch inactivating event. VAF of tumor initiating events below 1 indicate that the sequenced samples contained the expanded clone as well as the adjacent normal tissue.”*

c. I wonder what is the confidence interval for the allele frequency analysis: in other words, how much the variability of sequencing coverage at a given locus can influence allele

frequency calculation. This could be estimated by looking at the variability of estimated alleles frequencies for inherited transposons.

As suggested by the referee we explored the influence of sequencing coverage variability on allele frequency calculations. To do so we used germline heterozygous insertions to estimate the influence of the sequencing coverage on the variant allele frequency calculation. We have created approximately equal-sized bins based on coverage and plotted the variant allele frequencies for insertions within each bin. Firstly, this analysis showed that, as expected, for autosomal germline TE insertions VAF estimates show a median value of 0.58 (mean of 0.5). Secondly, sequencing coverage had very little impact on the calculated variant allele frequencies.

We then performed the same analysis for the somatic TE insertions. As expected, somatic VAFs were lower than the germline ones, reflecting varying fractions of cells carrying an insertion. For somatic insertions coverage appeared to have more influence on the estimate VAFs, however this effect was not dramatic and could likely be explained by much lower numbers of insertions per given coverage range than for the germline insertions.

Overall, our variant allele frequency estimates are not significantly affected by the coverage at the genomic position. For this reason, we opted for leaving the Fig 3B unchanged. In any case, as already mentioned above, we do not draw any strong conclusions on the precise timing of

the somatic TE insertions. Instead the VAF analysis supports our conclusion that insertions can arise both pre- and post-clonal expansion.

4. With regards to enrichment analysis, each family of transposon is likely to behave differently in terms of integration preference and selective pressure. Therefore, performing the analysis separately for each (abundant enough) family may reveal more insightful than pooling all insertions from all families.

We do agree with the referee that insertions of different TE families could show distinct enrichments in the genome and we have now made changes to account for this point. For this reason, we had already analyzed enrichment in chromatin features (Fig 6D and E) separately for the three major mobile TE families (*rover-LTR*, *copia-LTR* and *I-element*). We have now done the same for enrichments in genic vs non-genic regions (**new Fig 6B**) and presented the data separately for *rover-LTR* and *copia-LTR*. *I-element* did not show any significant enrichments/depletions likely due to low number of insertions. Similarly, for singleton insertions identified with the long-read sequencing, we have replaced the graph representing enrichments of all insertions in genic/non-genic regions, with one showing only *rover* insertions (**new Fig EV4**).

In the Results section, we have changed the text accordingly:

(p. 12) *“Although they were depleted from coding sequences, somatic LTR-element integration sites were significantly enriched in introns (rover and copia insertions) and 3’UTRs (rover insertions) (Fig 6B). Similar genic enrichment was evident in the singleton insertions of rover-LTR identified by the long-read sequencing of normal gut DNA pools (Fig EV4) (...)*

5. Expression is a pre-requisite for retrotransposition. Thus, the absence of expression for some retrotransposon families that exhibit high numbers of de novo insertions can only be explained by expression in a very restricted subset of cells of the intestine (discussed), or by the fact that transposition occurred much earlier during the development (not discussed). Please add a caveat in the discussion.

We thank the referee for raising this point. We have now added a caveat to the discussion.

(p.15) *“Alternatively, mobility might happen during earlier developmental stages and coincide with higher levels of transcription occurring at that time.”*

Also, if this is a matter of timing, do families with high somatic copy number and low expression (such as *rover*) more frequently exhibit high allele frequency suggestive of transposition in earlier stage of development, and vice-versa for families with high expression (such as *copia*)?

Our data do not support a simple relationship with timing and expression levels: in Fig 3C, we show that neither *rover* (lowly expressed in the adult gut) nor *copia* (highly expressed) insertions occurred significantly more frequently before or after *Notch* inactivating events, suggesting that both families were active to a similar extent pre- and post-clonally.

As the referee suggested, we have now also directly compared allele frequency values (rather than the pre- or post-clonal status) for all *rover*, *copia* and *I-element* insertions. Here *rover* autosomal insertions have significantly higher VAFs than *copia* insertions. These could indeed be consistent with *rover* being more active earlier in development where it may be more highly expressed than in the adult gut. However, this difference was not very high (median=0.42 for *rover*, median=0.32 for *copia*, p value= 4×10^{-4}) and not significant for the sex chromosome insertions. Keeping in mind that, as discussed above, VAFs values can only be used as estimates of the insertion timing, we do not consider this difference significant enough to merit its including in Fig. 3 and discussing in the manuscript text.

Finally, we would like to stress that even if we consider only insertions with allele frequency equal to or lower than the *Notch* inactivating event (and thus occurring in the adult gut), *rover*, regardless of its very low transcript levels, still remains by far the most active family. This strongly supports our conclusions on the lack of direct correlation between TE expression and mobility in the adult gut.

Minor points:

6. Please, add the orientation of transposons in the supplemental tables to provide a more complete description of the discovered insertions.

Insert orientation mapping was not implemented in our pipeline for calling insertions from the short-read Illumina data, but this information was recovered for the long-read ONT sequencing calls. We have now provided a **new Table EV4** with detailed characteristics of all putative somatic "singleton" insertions identified with the ONT sequencing, including insert orientation information. Cases of both forward and reverse insert orientation can be found in the dataset.

Additionally, orientation of *Notch* insertions from the short-read data is provided in **Fig 1E** and **new Appendix Fig S1**.

7. The enrichment of insertions in given regions of the genome is driven by the preference of the integration complexes for these regions, but also by natural selection, through cell death or cell growth, both for expanded clones and for bulk tissues. This is clearly discussed at the end of the manuscript, however some wording throughout the main text remains ambiguous and implicitly suggests that the observed biased reflects a biased integration process, e.g. 'preferential landing sites' (p.12), 'to further probe into integration preferences', etc. Please rephrase.

We thank the referee for careful reading and we agree that statements suggesting biased integration site selection should be avoided in the context of our study. We have now altered these in the text to be neutral using "distribution" or "genomic features" instead of "preference".

8. In samples where transposons are suspected to be causative for Notch inactivation, RNA-seq has not been performed - it is clearly challenging since it would have required to extract RNA from the same sample. Thus, it is unclear how the insertions actually impair Notch expression, and if they do. In principle, it is possible that other somatic mutations outside of Notch or in non-coding regions could phenocopy Notch inactivation. Please add a caveat and rephrase sentences such as 'we uncovered TE insertions causative for Notch inactivation' (p. 8).

We agree with this caveat of our study and we have now changed these to "likely causative".

We have also added the following caveat to the results:

(p. 4) *"Due to very limited sample material, we could not perform simultaneous RNA expression analysis in order to directly demonstrate the effect of TE inserts on Notch expression. However, as we did not detect evidence of other genetic alteration of Notch or Notch pathway components, we conclude that the TE insertions were most likely causative of the clonal expansion and Notch mutant phenotype."*

9. The fact that Notch gene is carried on the X-chromosome is only mentioned in the discussion. Please make it explicit in the introduction and results since it is essential to understand that a single mutational hit in males is sufficient to drive a loss-of-function.

Indeed, this lack of clarity about our experimental setup was also pointed out by the referee #1. To clarify this we have now:

- Added the following explanation at the beginning of the results section:

(p. 4) *"Since Notch is located on the X-chromosome and as such is present in a single copy in males, a single hit can lead to its inactivation (Fig 1A). In contrast, females, harboring two copies of Notch, do not or very rarely develop similar spontaneous Notch inactivation events."*

- Added to Fig1A a schematic of *Notch* localized on a single X chromosome in XY males and its loss in neoplastic clones

10. Showing reads as pairs in IGV screenshots (Fig. 1 and S1) would show more explicitly the reads supporting the empty allele. For the singleton mates, is there a meaning for the different colors?

We have now changed the IGV screenshots from Fig 1 and new Fig S2 (previously S1) to display sequencing reads as pairs.

Regarding the different colors for reads supporting insertions, they reflect the way those reads are tagged in our bioinformatic pipeline. Since reads are mapped to all *Drosophila* reference TEs, rather than to canonical sequences only, multiple colors at one insertion site mark homology to different insertions of the same TE family. In the final calls we then report the TE family (*rover*, *copia*, etc) of an insertion based on the full assembly of these reads.

The following explanation was already present in the figure legend:

“Reads supporting the TE insertion are colored according to homology to a specific TE insertion sequence. Multiple colors at a putative insert site frequently indicate homology to different reference copies of the same TE family”

We have now added a similar sentence of clarification to the Methods section (p. 22).

11. Fig. 1C-D, why only rover insertions were validated, and only a single one Sanger sequenced? This is important to support the sequencing approach.

Yes, we agree with the reviewer that ideally all samples would be validated through Sanger sequencing. However, for our clonal gut samples only very limited material is available for DNA isolation. We used 0.6 ng of gDNA for whole-genome sequencing library preparation (NexteraXT protocol), after which we typically had around 1-3 ng of DNA left for potential validations. However, we have now used this limiting DNA material and successfully expanded our PCR validation to all five *Notch* insertions from Fig 1C. These include a new full-length PCR validation of the *copia* insertion from sample D5 (**revised Fig 1D**) as well as a new 3' PCR validation of the *accord2* insertion in sample P51 (**new Appendix Fig S1**). The latter insertion showed only one-sided read support in the sequencing and, in agreement with this, we were unable to amplify this insertion with a full-length PCR. We think it is possible that this insert was associated with a structural variant of unknown nature, which rearranged the *Notch* sequence around the insertion site, hampering the design of specific oligos. Finally, we have now also sequence validated by Sanger sequencing the following insertions (**new Appendix Fig. S1**):

- sample D5 – *copia* – reverse orientation, full length sequence validation
- sample P15 – *rover* in 5'UTR – reverse orientation, partial sequence validation
- sample P51 – *accord2* – 3' breakpoint sequence validation

12. Fig. 2, it would be of interest to know the distribution of insertion per family and sample: are the active transposons mobilized in all samples equally or highly active in some samples, but inactive in others?

We thank the referee for this very valid question. We have now provided a per sample classification of active TE families as **Expanded View Figure EV1**. This analysis shows that there are no striking differences in mobile TE families between samples derived from the same genetic background.

We have added the following statement in the text:

(p. 6) “There were no striking differences in mobile TE families between samples of the same genetic background, suggesting that active TEs did not differ greatly between individuals (Fig EV1). In contrast, differences in mobile TE families were found between the two genotypes.”

13. Fig. 2E and 5C, where is (are) the break point(s) in the target motifs?

In fact, the exact breakpoint cannot be assigned to the target motif. As specified in the methods section we performed the target motif search analysis on sequences surrounding insertion sites (10 flanking nucleotides upstream and downstream of each insertion). The identified target motif can be present in any position within the searched sequence. It is true that the motif often overlaps with the TSD site, which is the site of insertion. However, it does not always center around the TSD and in some rare cases was even found outside of the TSD.

We have now clarified this in the Fig 2E legend as well as the results section by changing all statements “motif at *rover* insertion sites” into “motif around (+/- 10bp) *rover* insertion sites.

As an example, the screenshot below represents a snapshot of chromosome 2L *rover* insertion sites from the short-read sequencing samples. Black lines illustrated the searched sites (with a TE insertion site in the middle) and the red bars show the position of identified AT-rich motifs.

MEME analysis of somatic *rover* insertion sites from the short-read data (screenshot)

14. In Fig.6 and similar figures, were the p-values adjusted for multiple comparisons? (some look very high)

We thank the referee for this suggestion. We have now adjusted the p-values for multiple comparisons using a Benjamini-Hochberg correction method. However, this had only mild impact on the p-values. With a cutoff of p-value<0.001 our conclusions regarding enrichments and depletions of somatic insertions sites remain unchanged.

15. It is assumed that the differences between *Pros>2xGFP* and *DI>nlsGFP* strains are strain-specific, but could they also reflect tissue-specificities? Please clarify.

We strongly feel that the most likely explanation of inter-strain differences in somatic mobility is attributed to the genetic background. However, formally we cannot exclude that our detection of somatic insertions was partially biased by the experimental setup. More precisely, we selected clones for sequencing based on the accumulation of either GFP+ enteroendocrine cells (*Pros>2xGFP*) or GFP+ intestinal stem cells (*Delta>nlsGFP*). Typically *Notch* loss-of-function clones accumulate both cell types, though with different ratios. Thus, it is formally possible that we favored the selection of more EE-rich clones in the *Pros>2xGFP* males and more ISC-rich clones in the *Delta>nlsGFP* males. This could perhaps introduce some cell-types specific bias, but it is difficult to imagine that it could account for such an important disparity between the two strains. Additionally, such bias would only effect sub-clonal insertions (with low VAFs) and not insertions already present in the gut stem cells and clonally amplified upon the loss of *Notch* or insertions within *Notch*. As strain differences can already be appreciated only with the insertions in *Notch* and even more with pre-*Notch* insertions, we do not think that differences in cellular composition could explain the differences of mobile TE families between the two sets of samples.

Nevertheless, to acknowledge such possibility, we have now added the following text to the Results section:

(p. 6) *“However, we cannot exclude that some observed differences in mobility may have resulted from the differences in cell type-specific clone composition between the two genotypes, with enrichment of enteroendocrine cells (ProsperoGal4 driven GFP) or intestinal stem cells (DeltaGal4 driven GFP).”*

Thank you for submitting a revised version of your manuscript. Your study has now been seen by both original referees, who find that their main concerns have been addressed and recommend publication of the manuscript. Therefore, I would like to invite you to address the remaining minor referee comments and the following editorial issues.

Referee #1:

This is a lovely manuscript that reports exciting findings. I was already quite enthusiastic about the earlier version and raised mainly critiques of the text that were minor in scope. The authors have done a nice job of responding, with answers to my queries where that was warranted and with improvements to the text where that was needed. I have no further issues to raise.

Referee #2:

The authors have done a careful job answering all my previous concerns. The revisions have significantly improved the manuscript and it is now recommended for publication in the EMBO Journal. I have noted a few remaining minor points that could be corrected in the final version, to the discretion of the authors:

- in Fig EV5, middle panel, the top significant hits appear unlabelled
- I do not agree with this statement (p. 6) "There were no striking differences in mobile TE families between samples of the same genetic background, suggesting that active TEs did not differ greatly between individuals (Fig EV1)". Some samples do seem to exhibit very high levels of transposition (such as P3, P5 or P37), especially for the rover element. This is somehow reminiscent of the findings obtained from the Pan-cancer genome project in humans (Rodriguez-Martin et al, Nat Genet 2020), in which a few tumors also showed extremely high levels of L1 mobilization.

Response to referees:

Referee #1:

This is a lovely manuscript that reports exciting findings. I was already quite enthusiastic about the earlier version and raised mainly critiques of the text that were minor in scope. The authors have done a nice job of responding, with answers to my queries where that was warranted and with improvements to the text where that was needed. I have no further issues to raise.

We thank the referee for his suggestions to improve the text of our manuscript.

Referee #2:

The authors have done a careful job answering all my previous concerns. The revisions have significantly improved the manuscript and it is now recommended for publication in the EMBO Journal. I have noted a few remaining minor points that could be corrected in the final version, to the discretion of the authors:

- in Fig EV5, middle panel, the top significant hits appear unlabelled
- I do not agree with this statement (p. 6) "There were no striking differences in mobile TE families between samples of the same genetic background, suggesting that active TEs did not differ greatly between individuals (Fig EV1)". Some samples do seem to exhibit very high levels of transposition (such as P3, P5 or P37), especially for the rover element. This is somehow reminiscent of the findings obtained from the Pan-cancer genome project in humans (Rodriguez-Martin et al, Nat Genet 2020), in which a few tumors also showed extremely high levels of L1 mobilization.

We thank the referee for careful reading of the revised manuscript. We have now:

- Corrected Fig EV5 to include all labels of the significant hits
- Changed the sentence on p. 6 to:

"Although we observed varying levels of transposition, there were no striking differences in the types of mobile TEs between samples of the same genetic background, suggesting that active TEs did not differ greatly between individuals (Fig EV1)"

The authors performed the requested editorial changes.

Editor accepted the revised manuscript.

Corresponding Author Name: Katarzyna Siudeja, Allison Bardin

Journal Submitted to: EMBO J

Manuscript Number: EMBOJ-2020-106388